# Vaccinia virus hijacks ESCRT-mediated multivesicular body formation for virus egress

Moona Huttunen[1,2], Jerzy Samolej[2], Robert J Evans[2,3], Artur Yakimovich[1], Ian J White[1], Janos Kriston-Vizi[1], Juan Martin-Serrano[4], Wesley I Sundquist[5], Eva-Maria Frickel[2], Jason Mercer[1,2]

**Poxvirus egress is a complex process whereby cytoplasmic single membrane–bound virions are wrapped in a cell-derived double membrane. These triple-membrane particles, termed intracellular enveloped virions (IEVs), are released from infected cells by fusion. Whereas the wrapping double membrane is thought to be derived from virus-modified trans-Golgi or early endosomal cisternae, the cellular factors that regulate virus wrapping remain largely undefined. To identify cell factors required for this process the prototypic poxvirus, vaccinia virus (VACV), was subjected to an RNAi screen directed against cellular membrane-trafficking proteins. Focusing on the endosomal sorting complexes required for transport (ESCRT), we demonstrate that ESCRT-III and VPS4 are required for packaging of virus into multivesicular bodies (MVBs). EM-based characterization of MVB-IEVs showed that they account for half of IEV production indicating that MVBs are a second major source of VACV wrapping membrane. These data support a model whereby, in addition to cisternae-based wrapping, VACV hijacks ESCRT-mediated MVB formation to facilitate virus egress and spread.**

## Introduction

All enveloped viruses must modulate cellular membrane systems throughout their lifecycles, from hijacking endosomes for entry and diverting cellular membrane compartments for replication, to taking over cellular secretion systems for viral egress and immune evasion (Welsch et al, 2007; Mercer et al, 2010; Glingston et al, 2019). As exclusively cytoplasmic replicating viruses, poxviruses are no exception to this rule (Moss, 2007). Moreover, as these viruses produce two distinct infectious virion forms—with multiple membranes sequestered from different cellular membrane compartments—poxviruses are perhaps more reliant on modulating cellular membrane compartments than most other enveloped viruses.

The formation of these distinct infectious virus forms termed intracellular mature virions (IMVs) and extracellular enveloped virions (EEVs) begins with the assembly of single-membrane IMVs in viral factories (Moss, 2007). Once formed, a subset of IMVs is transported to designated wrapping sites where they acquire two additional virus-modified membranes (Smith et al, 2002; Condit et al, 2006; Moss, 2007; Roberts & Smith, 2008). These triple-membrane virions termed intracellular enveloped virions (IEVs) are then transported to the cell surface where they undergo fusion. Leaving behind their outermost membrane, these double-membrane virions remain as cell-associated enveloped virions (CEVs) or are released to become EEVs.

The wrapping membranes contain nine viral proteins not found in IMVs, some of which direct the processes of wrapping and cell surface transport (Smith et al, 2002; Roberts & Smith, 2008). The extracellular virion (EV) membranes themselves have been reported to be derived from trans-Golgi (TGN) and/or endosomal cisternae (Smith et al, 2002; Condit et al, 2006; Moss, 2007; Roberts & Smith, 2008). However, the cellular machinery involved in IMV wrapping/IEV formation, is largely unknown.

It is generally accepted that many enveloped viruses exit cells by budding from the plasma membrane (Welsch et al, 2007). To achieve this, numerous viruses rely on cellular endosomal sorting complexes required for transport (ESCRT) machinery, which mediates reverse topology cellular fusion events such as the formation of intraluminal vesicles (ILVs) and cytokinesis (McCullough et al, 2013; Scourfield & Martin-Serrano, 2017; Vietri et al, 2020). ESCRT includes five core complexes required for recognition, clustering and nucleation of ubiquitinated cargoes (ESCRT-0, ESCRT-I, and ESCRT-II), filament formation for membrane pinching (ESCRT-III), and an AAA + ATPase for the scission, and for removing ESCRT complexes after fusion (VPS4) (McCullough et al, 2013; Vietri et al, 2020).

Interestingly, it has been reported that depletion of TSG101, a component of ESCRT-I, or ALG-2 interacting protein X (ALIX)—an ESCRT accessory protein—reduces EEV formation (Honeychurch et al, 2007). Intrigued by our identification of TSG101 and VPS4 in a high-throughput RNAi screen for cell factors that impact virus spread, we

---

[1]Medical Research Council-Laboratory for Molecular Cell Biology, University College London, London, UK [2]Institute of Microbiology and Infection, University of Birmingham, Birmingham, UK [3]Host-Toxoplasma Interaction Laboratory, The Francis Crick Institute, London, UK [4]Department of Infectious Diseases, King's College London, London, UK [5]Department of Biochemistry, University of Utah, Salt Lake City, UT, USA

Correspondence: Moona.Huttunen@utu.fi; J.P.Mercer@bham.ac.uk

 

decided to pursue the role of ESCRT machinery during VACV infection. This led to the identification of ESCRT-mediated VACV wrapping and the discovery that multivesicular bodies (MVBs) serve as a major non-cisternae membrane source for the formation of IEVs.

## Results

### Membrane-trafficking RNAi screen implicates ESCRT machinery in VACV spread

To identify novel cell factors used by VACV to facilitate virus spread, we developed a fluorescent plaque formation assay compatible with RNAi screening (Fig 1A). For this we used a virus (VACV E-EGFP L-mCherry) that expresses EGFP and mCherry under the control of an early or late viral promoter, respectively. HeLa cells were treated with RNAi targeting various membrane-trafficking proteins for 72 h (Fig 1A and Table S1). Cells were then infected with VACV E-EGFP L-mCherry at MOI 0.02, and at 8 hours postinfection (hpi) viral DNA replication and late gene expression were inhibited using cytosine arabinoside (AraC) (Furth & Cohen, 1968; Schabel, 1968), after which infection was allowed to proceed for a further 16 h. This workflow resulted in fluorescent plaques containing magenta primary infected cells surrounded by green secondary infected cells (Fig 1A). Cells were stained for nuclei, imaged and quantified for cell number and infection markers (mCherry and EGFP). Quantification of the number of primary (magenta) and secondary infected (green) cells allowed us to differentiate between defects in primary infection by IMVs and defects in virus spread which could be caused by attenuated virion formation or entry of CEVs or EEVs into surround cells. Of the 224 cell factors screened, 42 reduced VACV spread by 25% or more (Fig 1B and Table S1). Hits were assigned to functional annotation clusters using DAVID (Huang et al, 2009a, 2009b). Protein–protein interactions within and between clusters were mapped using STRING (Szklarczyk et al, 2019) (Fig 1B). Enriched functional clusters included Rab GTPases, endocytosis related proteins (actin and clathrin), and SNAP-SNARE receptors.

The identification of actin and clathrin regulatory proteins is consistent with their role in actin tail formation during VACV infection (Leite & Way, 2015). The large number of Rab GTPases and set of SNAP-SNARE proteins identified is also in line with the role of retrograde trafficking for the recycling and transport of wrapping proteins from the plasma membrane to the Golgi (Harrison et al, 2016; Sivan et al, 2016).

Two additional small clusters were identified, autophagy and ESCRT machineries (Fig 1B). The autophagy cluster is composed of ATG12, Beclin 1, and LC3A, of which ATG12 was implicated in virus-mediated inhibition of autophagy (Moloughney et al, 2011). The ESCRT cluster contains ESCRT machinery proteins (TSG101 and VPS4A) and a protein related to ESCRT machinery function (NEDD4) (Fig 1B). It has been shown that depletion of TSG101 or the accessory protein ALIX reduce EEV yield when HeLa cells are infected with VACV (Honeychurch et al, 2007). Our screen showed that RNAi-mediated depletion of TSG101, VPS4A, and NEDD4 caused a reduction of 39%, 60%, and 34% in virus spread (Table S1), and a concomitant reduction in VACV plaque formation (Fig 1C). As a

validation step, we depleted TSG101 or ALIX and determined IMV and EEV yields 24 hpi. Immunoblot analysis indicated that TSG101 and ALIX were reduced by 77% and 96%, respectively (Fig 1D). Whereas IMV yields were unaffected by loss of either protein (Fig 1E), EEV yields were reduced by 25% upon knockdown of TSG101, and 41% upon knockdown of ALIX (Fig 1F).

Collectively, the RNAi screen uncovered a subset of cellular factors and processes required for VACV spread. The identification and validation of ESCRT components is consistent with the reported role of TSG101 and ALIX in VACV EEV formation (Honeychurch et al, 2007). As mechanistic understanding of the role of ESCRT machinery in poxvirus wrapping and/or egress is lacking, we chose to focus on determining the role(s) of ESCRT complexes during VACV infection.

### VPS4B is a pro-viral factor required for VACV EEV formation

Mammalian cells express two forms of the ESCRT ATPase: VPS4A and VPS4B (Scheuring et al, 2001). Having identified VPS4A AAA ATPase in the screen we depleted VPS4A and VPS4B individually to further investigate the role of ESCRT machinery in VACV infection. Initial qRT-PCR validation of VPS4A and VPS4B depletion over 72 h (as used in the siRNA screen) indicated that the expression levels of both VPS4A and VPS4B were reduced with siRNA targeting either protein (Fig 2A). By optimizing the knockdown conditions, we achieved VPS4A and VPS4B knockdown efficiency of >99% with their respective siRNAs (Fig 2B). Although depletion of VPS4A resulted in a 2.5-fold increase in VPS4B mRNA and depletion of VPS4B reduced VPS4A mRNA by 25% relative to control cells (Fig 2B), the complete absence of VPS4A and VPS4B under these conditions provided the opportunity to tease apart the requirement of the proteins in virus spread.

Using the spread screen assay using these knockdown conditions, we found that loss of VPS4B, but not VPS4A, reduced virus spread (Fig 2C). Analysis of primary and secondary infection showed that depletion of VPS4A did not impact primary infection but was highly variable with regard to secondary infection, whereas depletion of VPS4B reduced the number of secondary infected cells without impacting the number of primary infected cells relative to control siRNA (Fig 2C; inset).

These results suggested that VPS4B, but not VPS4A, might affect EV-related egress of VACV. To assess this, cells depleted for VPS4A or VPS4B were infected with VACV and the production of IMVs and EEVs assessed at 24 h (Fig 2D). Consistent with the spread assay (Fig 2C), loss of either VPS4A or VPS4B had no impact on IMV formation. On average, loss of VPS4A showed no defect in the number of EEVs produced, whereas EEVs were down 42% upon depletion of VPS4B (Fig 2C).

Given the role of ESCRT machinery in membrane budding and fission, we reasoned that VPS4B was either involved in the intracellular wrapping of virions or their budding and release at the cell surface. To differentiate between these two, cells depleted for VPS4A or VPS4B were infected with VACV containing a fluorescent core (Western Reserve [WR] A5-EGFP). At 8 hpi, cells were fixed and stained for the VACV envelope protein B5 which is required for virus wrapping (Engelstad & Smith, 1993) and in this assay, marks wrapping membranes, IEVs, and CEVs. As expected, in control cells, significant colocalization between core (A5) and EV membrane protein (B5) was seen at the cell periphery consistent with the formation of IEVs and CEVs (Fig 2E, scrambled siRNA [siSCR]).

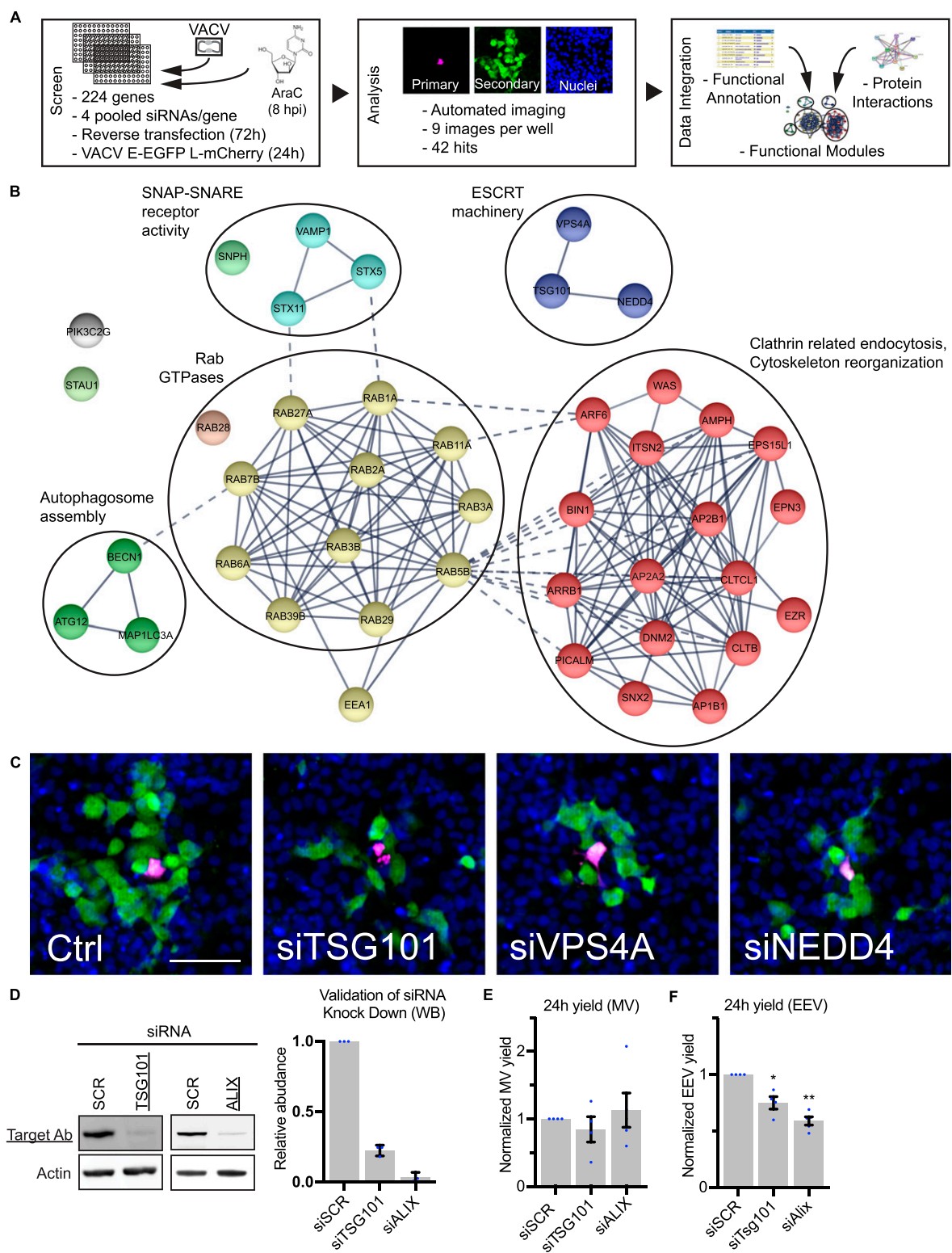

**Figure 1. RNAi screen targeting membrane-trafficking proteins reveals 42 EV egress factor candidates.**
**(A, B, C)** A siRNA screen targeting 224 cellular membrane-trafficking factors was used to identify host proteins required for EV related viral egress. siRNA-transfected HeLa cells were infected with a VACV (E– EGFP/L-mCh) and fractions of early and late infected cells were quantified. **(A)** Workflow and screening strategy are shown in (A). **(B)** Gene knockdowns that decreased late/early infection ratio ≥25% were selected as hits. The 42 hit candidate genes are shown grouped into functional modules. n = 2. See also Table S1. **(C)** Representative images of endosomal sorting complexes required for transport machinery hits. Blue = DAPI, magenta = late infection, green = early infection. Scale bar 200 μm. **(D)** Western Blot validations of TSG101 and ALIX siRNA knockdown. Mean ± SEM, n = 3. Statistical analysis was performed using unpaired two-tailed $t$ test (****$P$ < 0.0001). **(E, F)** 24-h intracellular mature virion and extracellular enveloped virion yields from control (siSCR), TSG101, and ALIX depleted cells. Data are mean ± SEM, n = 4, normalized to control (siSCR). Statistical analysis was performed using unpaired two-tailed $t$ tests (*$P$ < 0.05; **$P$ < 0.01).

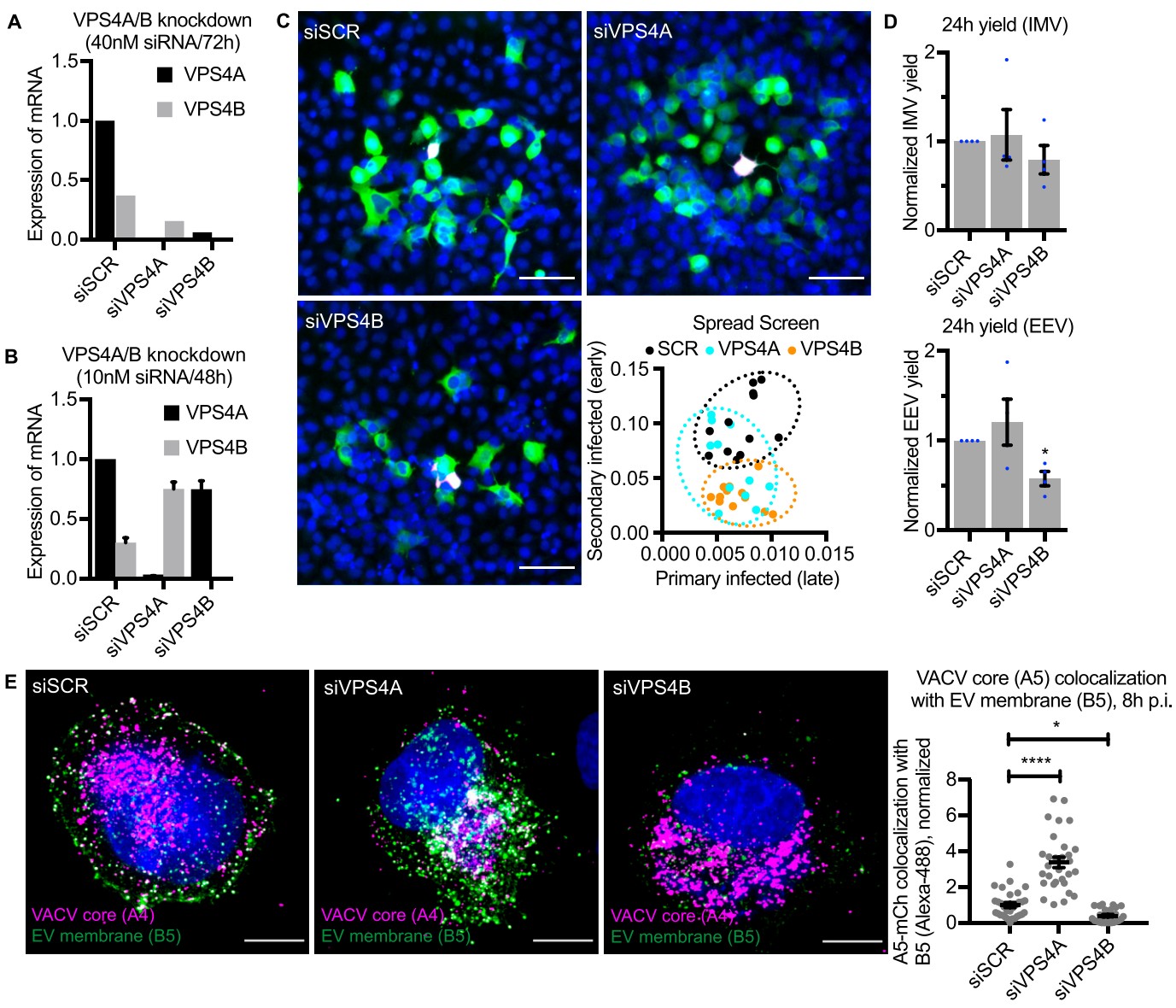

**Figure 2. VPS4 plays a role in intracellular mature virion wrapping and VACV spreading.**
**(A)** qRT-PCR validations of siRNA knockdown efficiencies for VPS4A/B (40 nM siRNA/72 h). Data are mean ± SEM, n = 3. **(B)** qRT-PCR validations of siRNA knockdown efficiencies for VPS4A/B (10 nM siRNA/2 × 24 h). Data are mean ± SEM, n = 3. **(C)** Representative images and quantification of EV related spreading in VPS4A/B depleted HeLa cells. Blue = DAPI, magenta = late infection, green = early infection. n = 2. Scale bars 50 μm. **(D)** 24-h intracellular mature virion and extracellular enveloped virion yields in control (siSCR) and VPS4A/B depleted cells. Data are mean ± SEM, n = 4, normalized to control (siSCR). Statistical analysis was performed using unpaired two-tailed *t* tests (*$P$ < 0.05). **(E)** Representative images (maximum intensity projections) of VACV (mCh-A4) infected VPS4 depleted HeLa cells at 8 hpi, and quantification of VACV core and EV membrane protein (B5) colocalization. Blue = DAPI, magenta = VACV core (A4), green = VACV envelope protein B5. Scale bars = 10 μm. Data are mean ± SEM, three repeats, n = 30 cells per condition. Statistical analysis was performed using unpaired, nonparametric Kolmogorov–Smirnov test (**$P$ < 0.01; ****$P$ < 0.0001).

Colocalization of these two signals appeared to increase in VPS4A-depleted cells with high colocalization within wrapping sites at the cell periphery (Fig 2E, vacuolar protein sorting-associated protein 4A siRNA [siVPS4A]). In contrast, loss of VPS4B appeared to cause a decrease in colocalization with core signal (A5) often appearing adjacent to the B5-positive wrapping membrane signal within cells (Fig 2E, siVPS4B). Quantification showed that relative to control cells, in VPS4A depleted cells colocalization between these two markers increased by threefold, whereas in VPS4B-depleted cells colocalization was decreased by 60% (Fig 2E; inset).

Upon VPS4B depletion, we did not see accumulation of virions at the cell periphery and little colocalization between core and IEV wrapping membrane. Knockdown of VPS4A, which increased expression of VPS4B, resulted in a complementary increase in core/wrapping membrane colocalization, suggesting that VPS4B is a proviral factor involved in VACV EEV formation.

## ESCRT-III complex contributes to VACV intracellular wrapping

The above results suggested that ESCRT machinery is involved in intracellular wrapping of newly assembled VACV virions (or IMVs). Following this hypothesis, we sought to investigate the involvement of ESCRT-III. To this end we depleted components of ESCRT-III (charged multivesicular body protein [CHMP]1A, CHMP1B, CHMP2B, CHMP3, CHMP4A, CHMP4B, CHMP4C, CHMP5, and CHMP6) using siRNA. Immunoblot or qRT-PCR analyses indicated that each factor was depleted by 50% or more (Fig 3A–C). As the original spread screen did not differentiate between reduced IMV or EEV production in primary infected cells, 24 h yields were used to assess the impact of ESCRT-III depletion on VACV IMV and EEV formation. No defect in IMV formation was seen upon loss of any ESCRT-III components tested (Fig 3D). Conversely, loss of CHMP1A, CHMP3, CHMP4C, and CHMP6 each reduced EEV yield by 25–44%, and depletion of CHMP4B and CHMP5 increased EEV yield by 95% and 44%, respectively (Fig 3E). These results indicated that in addition to VPS4, ALIX, and TSG101, multiple components of ESCRT-III are involved in VACV EEV formation.

As before, to determine the stage of EEV formation that ESCRT-III components are involved in, cells were depleted of CHMP1A, CHMP3, CHMP4B, CHMP4C, CHMP5, or CHMP6 and infected with WR A5-EGFP virus. At 8 hpi, cells were fixed and stained for the EV membrane marker B5. Control cells and those depleted of CHMP4B or CHMP5 displayed the expected distribution of core and EV markers, with colocalization occurring within perinuclear wrapping sites and in wrapped virions at the cell periphery (Fig 4A; see region of interests). Cells depleted of CHMP1A, CHMP3, CHMP4C, and CHMP6 showed a markedly different phenotype. With the exception of CHMP4C depletion, in which B5 distribution was dispersed, cells depleted of CHMP1A, CHMP3, and CHMP6 contained enlarged vesicular structures positive for the EV membrane marker B5. Strikingly, in these cells, multiple IMVs (indicated by core only marker; A5) appeared to be associated with the limiting membrane of these B5-positive vesicular structures (Fig 4A; see region of interests). The phenotypes were quantified using core/EV membrane marker colocalization as a proxy for IEV formation (Fig 4B). In line with the EEV yields, depletion of CHMP4B and CHMP5 showed a trend of increased colocalization, whereas knockdown of CHMP1A, CHMP3, CHMP4C, or CHMP6 resulted in 47%, 70%, 27%, and 72% less colocalization, respectively, when compared with control cells (Fig 4B). These results are in further support of ESCRT-mediated intracellular VACV wrapping.

## VACV virions associate with cisternae and MVBs within wrapping sites

VACV IMV wrapping takes place adjacent to viral factories in the area of the microtubule organizing center. It has been reported that wrapping cisternae are derived solely from the TGN (Hiller & Weber, 1985; Schmelz et al, 1994). This, however, is controversial as early endosomal cisternae and late endosomal structures have also been implicated in wrapping (Tooze et al, 1993; Husain & Moss, 2001; Chen et al, 2009), and Brefeldin A (BFA)—which collapses the TGN—decreases but does not fully attenuate EEV formation (Ulaeto et al, 1995). To get an overview of membranous organelles in the

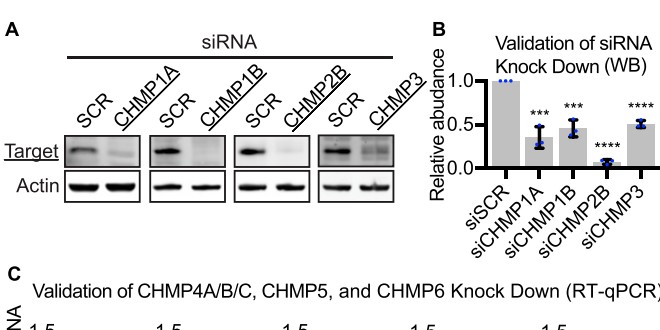

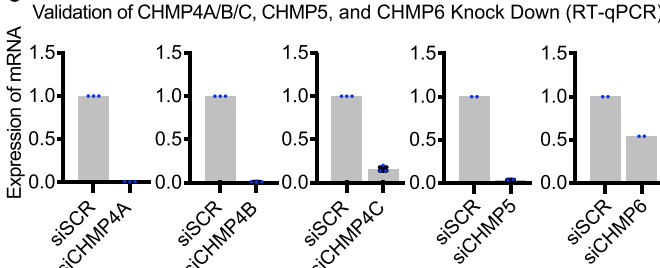

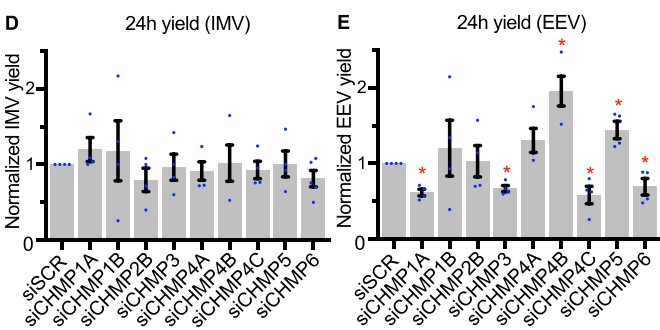

**Figure 3. Endosomal sorting complexes required for transport-III plays a role in extracellular enveloped virion formation.**
**(A, B)** Western Blot validations of CHMP protein siRNA knockdowns in HeLa cells. Data are mean ± SEM, n = 3, normalized to control (siSCR). Statistical analysis was performed using unpaired two-tailed $t$ tests (***$P$ < 0.001; ****$P$ < 0.0001).
**(C)** qRT-PCR validation of siRNA knockdown efficiencies for CHMP4A/B/C, CHMP5, and CHMP6 depletions in HeLa cells. Data are mean ± SEM, n = 3 for CHMP4A/B/C, and n = 2 for CHMP5 and CHMP6, normalized to control (siSCR). Statistical analysis was performed using unpaired two-tailed $t$ tests (****$P$ < 0.0001). **(D, E)** 24 h intracellular mature virion and extracellular enveloped virion yields in control (siSCR) and CHMP depleted cells. Data are mean ± SEM, n = 4, normalized to control (siSCR). Statistical analysis was performed using unpaired two-tailed $t$ tests (*$P$ < 0.05).

vicinity of cytosolic VACV wrapping sites, we infected cells with a virus that express a fluorescent version of the EV membrane protein F13. At 8 hpi, cells were fixed and stained for markers of early endosomes (EEA1), late endosomes (CD63), lysosomes (LAMP1), cis-Golgi (GM130), or TGN (TGN46). Quantification indicated that the VACV EV membrane protein F13 showed the greatest amount of overlap (38.6%) with the TGN, followed by late endosomes (28.5%), lysosomes (21.8%), cis-Golgi (17.3%), and early endosomes (14.2%) (Fig 5B).

As we noted a large variety of cellular membrane structures within the vicinity of virion wrapping sites, we used transmission electron microscopy (TEM) to further investigate their association with virions during wrapping (Fig 5C). An overview of infected cell 8 hpi shows that viral factories containing hallmarks of viral morphogenesis (crescents and immature virions) are largely devoid of cellular membrane structures (Fig 5C1'). Consistent with the

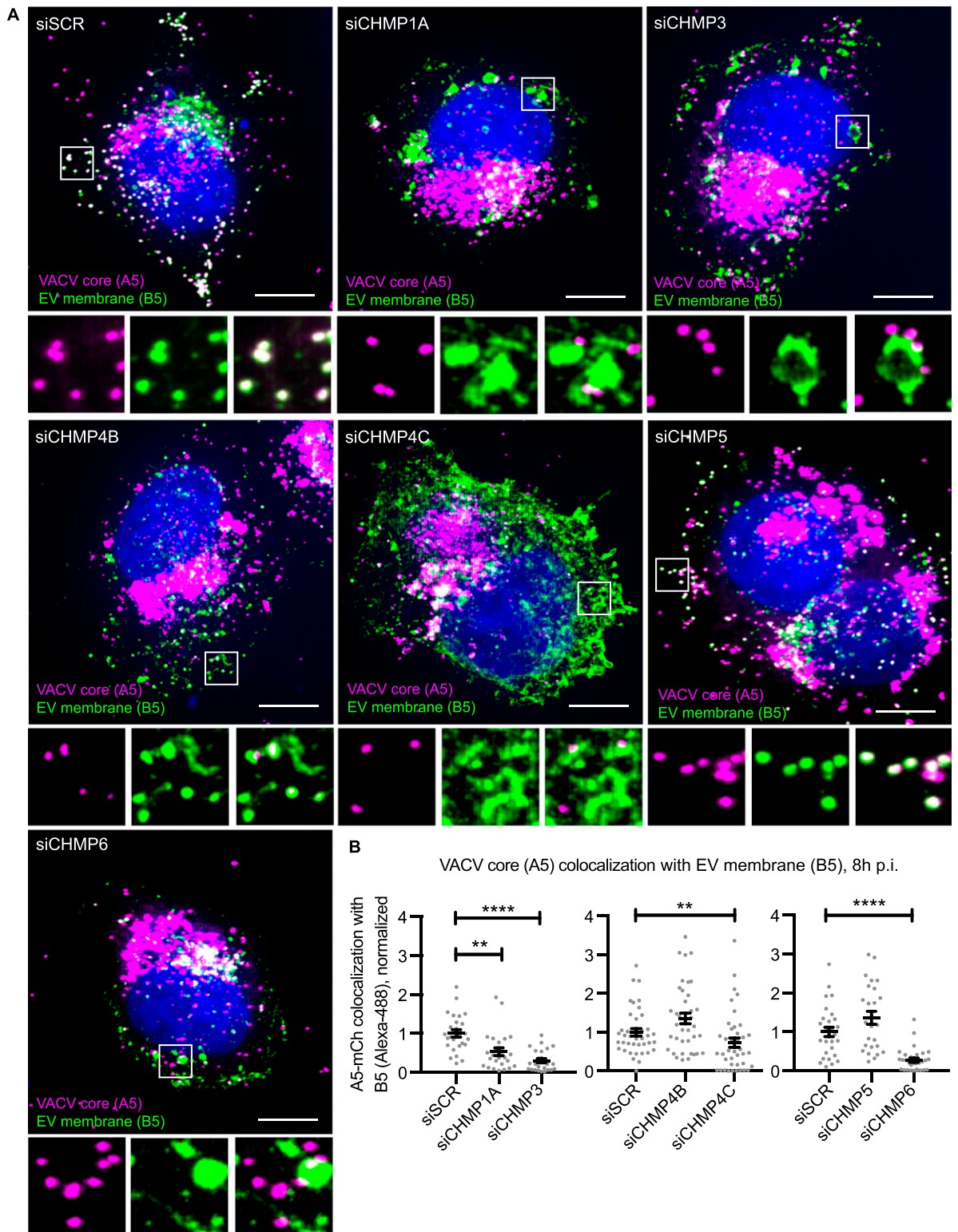

**Figure 4. Endosomal sorting complexes required for transport-III has a role in intracellular mature virion wrapping.**
**(A)** Representative images (maximum intensity projections) of VACV (mCh-A4)-infected CHMP-depleted HeLa cells 8 hpi. **(B)** Quantification of VACV core and EV membrane protein (B5) colocalization. Blue = DAPI, magenta = VACV core (A4), green = VACV envelope protein B5. Scale bars = 10 μm. Data are mean ± SEM, three repeats, n = 30 cells per condition. Statistical analysis was performed using unpaired, nonparametric Kolmogorov–Smirnov test (**$P < 0.01$; ****$P < 0.0001$).

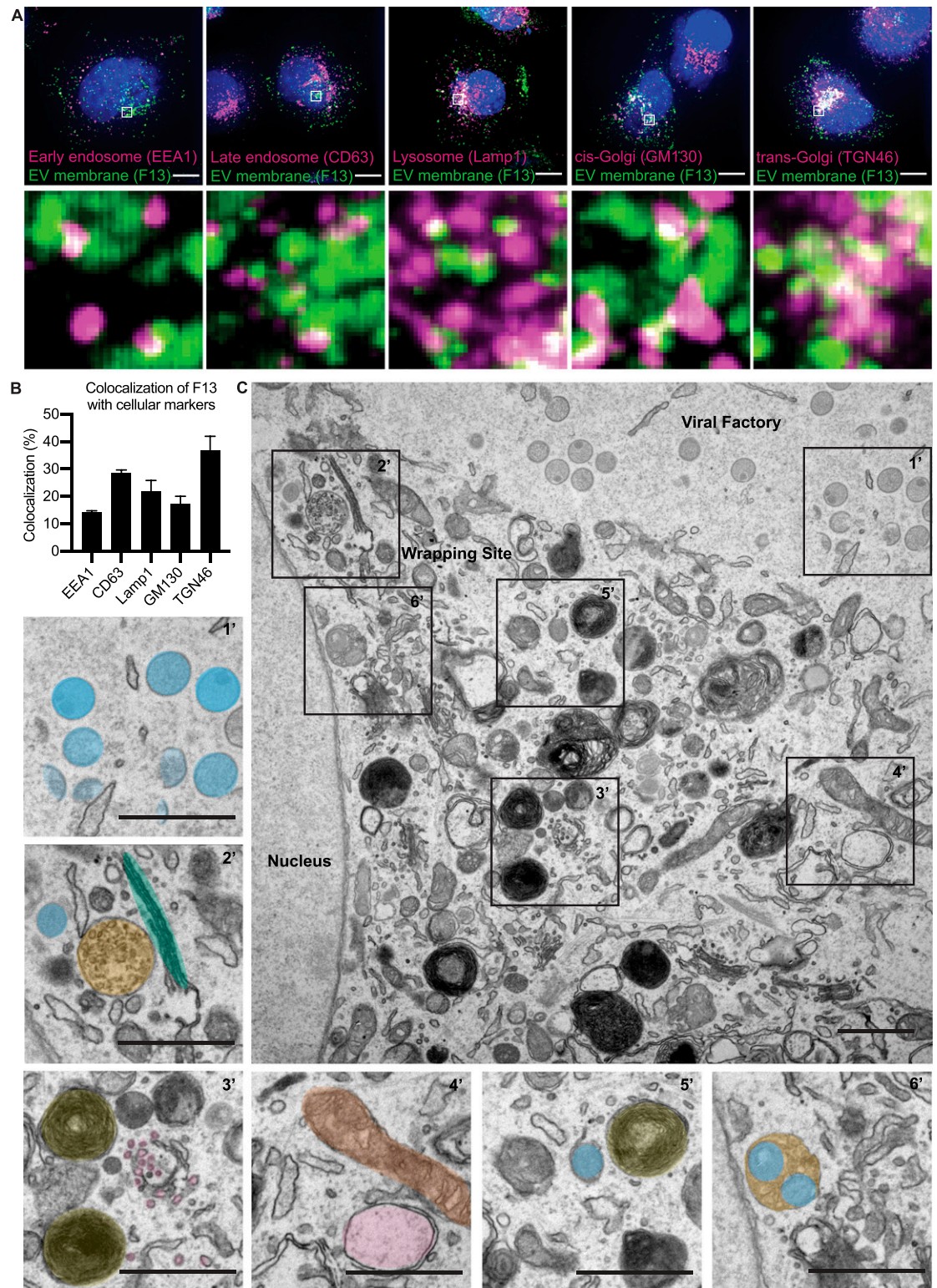

**Figure 5. MVBs serve as a second major source of VACV wrapping membrane.**
**(A)** Immunofluorescent (IF) imaging 8 hpi shows several cellular membrane markers in close proximity of EV membrane protein F13 (EEA1, CD63, Lamp1, GM130, and TGN46). Maximum intensity projections. Scale bars = 10 μm. Insets = 20× zoom. **(A, B)** Quantification of F13 colocalization with cell markers from (A). **(C)** EM imaging 8 hpi illustrates the location of viral replication site (viral factory) with different stages of mature virion morphogenesis (crescents and immature virions [1']). In addition, the imaging shows that in the areas of intracellular mature virion wrapping, various cellular membrane structures are in close proximity of wrapping virions. 2': blue = intracellular mature virion, green = Golgi stacks, orange = multivesicular body (MVB), 3': yellow = lysosomes, purple = small vesicles, 4': brown = mitochondria, pink = early endosome, 5': blue = wrapping virion, yellow = lysosome, 6': orange/blue = virions bud into MVB. Scale bars = 1 μm.

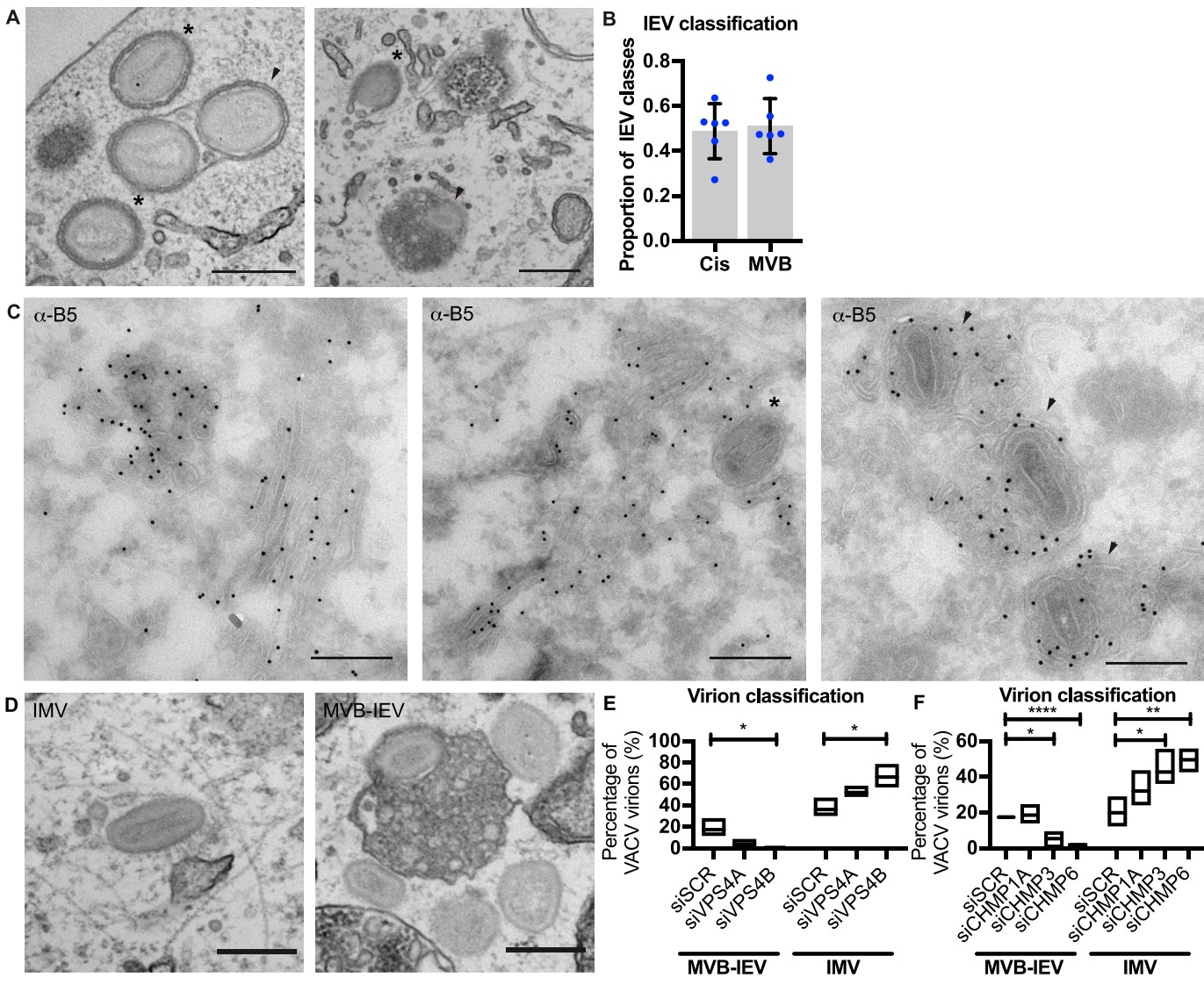

**Figure 6. Formation of MVB–intracellular enveloped virion (IEVs) is dependent on endosomal sorting complexes required for transport machinery.**
**(A)** Representative EM images of VACV-infected HeLa cells (8 hpi) showing cis-wrapping (asterisk), and MVB-IEVs (arrowheads). Scale bars = 500 nm. **(B)** Quantification of the IEV classes. Data are representative of six biological replicates. Data are mean ± SEM, representative of n = 80–100 counted viral particles per replicate. **(C)** Representative images of VACV-infected HeLa cell cryosections (8 hpi) where EV membrane protein B5 is immunolabelled with gold. Cisternae wrapped virions (asterisks) and MVB-IEVs (arrowheads). Scale bars = 500 nm. **(D, F)** Representative EM images of intracellular mature virion and MVB-IEV, and quantifications of proportions of these classes. Data are representative of three biological replicates. Data are mean with min and max values. Statistical analysis was performed using unpaired two-tailed $t$ tests (*$P$ < 0.05; **$P$ < 0.01; ****$P$ < 0.0001).

immunofluorescence (IF) imaging (Fig 5A), VACV wrapping sites—in addition to virions—were packed with various cellular membranes including Golgi stacks and MVBs (Fig 5C2'), late endosomes, lysosomes, small vesicles/tubules, early endosomes, and mitochondria (Fig 5C3' and C4'). Among all of these cellular organelles, we saw evidence of cisternae-based IMV wrapping (Fig 5C5') and we also found virions within MVBs (Fig 5C6').

### MVBs serve as VACV-wrapping organelles

Cisternae-based wrapping has been previously reported (Smith et al, 2002; Condit et al, 2006; Moss, 2007; Roberts & Smith, 2008);

as such in the literature VACV wrapping is "illustrated" as IMVs getting tightly wrapped within double-membrane sheets. However, our results suggest that IEVs can also be formed by the budding of IMVs into the lumen of MVBs (Figs 5B6' and 6A). To determine if MVB-based wrapping is a common event, the proportion of tightly wrapped cisternae-IEVs and MVB-IEVs was quantified using TEM at 8 hpi. We found that approximately equal numbers of cisternae-IEVs and MVB-IEVs, sometimes containing multiple virions, were formed in infected cells (Fig 6B).

We reasoned that as part of a productive VACV wrapping pathway that accounts for 50% of IEV formation, these MVB-IEV structures should contain EV membrane proteins. To assess this, immuno-EM directed against the wrapping membrane protein B5 was performed

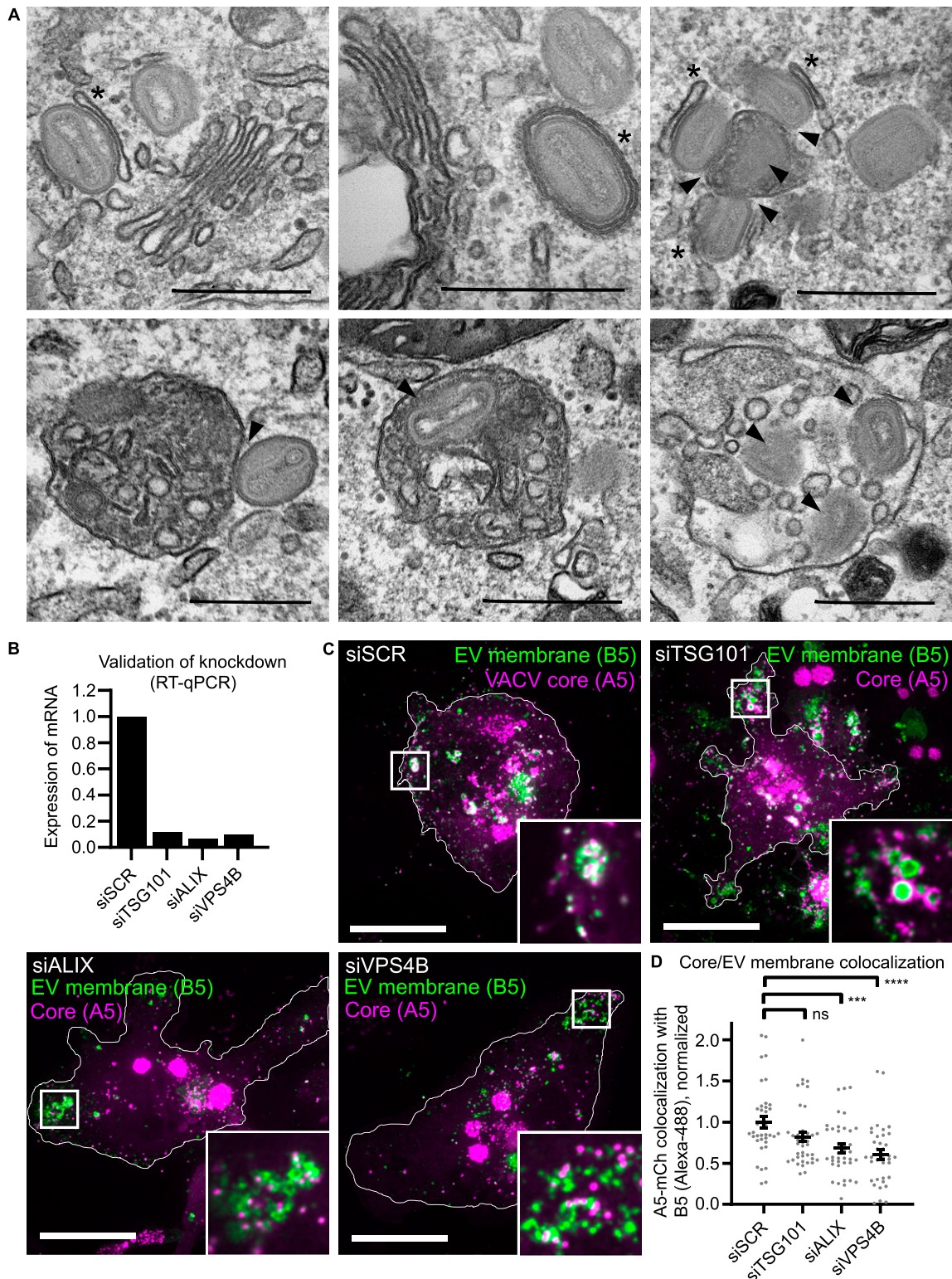

**Figure 7.  Endosomal sorting complexes required for transport–dependent formation of MVB-intracellular enveloped virions occurs in monocyte-derived macrophages.**
**(A)** Representative EM images of VACV infected THP-1 macrophages (24 hpi). Both cis-wrapping (indicated by asterisk), and MVB-intracellular enveloped virions (arrowheads) are present. Scale bars = 500 nm. **(B)** qRT-PCR validation of siRNA knockdown efficiencies of TSG101, ALIX, and VPS4B depletion in THP-1s. Data are average of duplicate samples normalized to control (siSCR). **(C)** Representative images (maximum intensity projections) of THP-1 macrophages depleted of the indicated protein and infected with VACV (mCh-A5), fixed at 24 hpi, and immunostained for B5. Magenta = VACV core (A5), green = VACV envelope protein B5. Scale bars = 10 μm. Insets = 8× zoom. **(C, D)** Quantification of VACV core and EV membrane protein (B5) colocalization from (C). Data are mean ± SEM, two biological replicates, n = 35–45 cells per condition. Statistical analysis was performed using unpaired, nonparametric Kolmogorov–Smirnov test (***P < 0.001; ****P < 0.0001).

on cryo-EM sections of infected cells. B5 was found in both cisternae and MVBs (Fig 6C; left). Cisternae in the process of wrapping IMVs were B5-positive (Fig 6C; middle), as were MVBs containing virions (Fig 6C; right). We noted that B5 was found on the limiting membrane of MVBs, on the intraluminal vesicles (ILVs) and on the inner EV membrane of virions within (Fig 6C; left and right). From this we concluded that the virions within these MVB-structures are IEVs.

### VACV MVB-IEV formation is ESCRT dependent

We have demonstrated that components of the ESCRT pathway, namely, VPS4B and multiple ESCRT-III components (CHMP1A, CHMP3, CHMP4C, and CHPM6) are required for IEV formation and virus spread. Complementing this, we have discovered that B5-positive MVBs appear to serve as a major source of membrane for VACV IMV wrapping. To link these two findings, we asked if MVB-IEV formation requires ESCRT machinery using quantitative TEM. For this, cells were depleted of VPS4A, VPS4B, CHMP1A, CHMP3, or CHMP6 and assessed for MVB-IEV and IMV formation at 8 hpi, as exemplified in Fig 6D. In control cells, 17% of virions could be classified as MVB-IEVs (Fig 6E; siSCR). When cells were depleted of VPS4A the portion of MVB-IEVs dropped to 4%, and when VPS4B was depleted, zero IEV-MVBs could be detected (Fig 6E). Further supporting the role of ESCRT machinery in IMV wrapping, a concomitant increase in the number of IMVs was seen upon VPS4A and B depletion: from 36% in control cells to 52% in the case of VPS4A and 67% in the case of VPS4B (Fig 6E). Quantification of MVB-IEVs and IMVs in CHMP1A-, CHMP3-, and CHMP6-depleted cells largely mirrored the VPS4 results. In control cells, 17% of virions were classified as MVB-IEVs (Fig 6F). Whereas CHMP1A depletion showed no significant difference, the proportion of MVB-IEVs decreased to 5% upon CMHP3 depletion and to 2% upon depletion of CHMP6 (Fig 6F). Again, the proportion of IMVs increased in cases where MVB-IEVs decreased: from 20% in the control cells to 42% and 49% in CHMP3 and CHMP6 depleted cells, respectively. These results demonstrate that ESCRT-dependent, MVB-based wrapping is a major source of VACV IEV formation, and whose loss results in the accumulation of IMVs that have failed to be wrapped and exocytosed and a concomitant reduction in EEVs released from cells.

### VACV MVB-IEV formation occurs in monocyte-derived macrophages

It has been reported that human monocyte-derived macrophages produce large amounts of EEVs, suggesting that macrophages may represent a major cell-source for virus dissemination in vivo (Byrd et al, 2014). To investigate IEV formation in THP-1 macrophages, THP-1 monocytes were differentiated into macrophages and subsequently infected with VACV. At 24 hpi, cells were processed for TEM and VACV wrapping investigated. As in HeLa cells, we observed a large variety of cellular membrane structures within wrapping sites. In addition to classic golgi cisternae-based VACV wrapping, we observed IMVs in association with MVBs, as well as single and multi-virion MVB-IEVs (Fig 7A).

### ESCRT contributes to MVB-IEV formation in THP-1 macrophages

To assess the impact of ESCRT depletion on VACV MVB-IEV formation in macrophages, THP-1 monocytes were transfected with siRNAs directed against ESCRT components TSG101, ALIX, or VPS4B before differentiation. qRT-PCR showed that each protein was depleted by >5-fold relative to control siRNA (Fig 7B). Control cells and those depleted for TSG101, ALIX, or VPS4B were infected with WR A5-mCherry, fixed and stained for EV envelope protein B5 at 24 hpi. As before, phenotypes were quantified using core/EV membrane marker colocalization as a proxy for IEV formation. In THP-1 cells, significant colocalization between core (A5) and EV membrane (B5) was seen in individual IEVs, as well as in large intracellular structures consistent with MVB-IEV formation (Fig 7C, siSCR). Similar to what we observed in HeLa cells upon loss of VPS4B and several CHIMPs (Figs 2D and 4B), loss of TSG101, ALIX and VPS4B caused a decrease in colocalization between core and EV membrane signals, with cores often appearing adjacent to enlarged B5-positive wrapping membranes (Fig 7C, siTSG101, siALIX, and siVPS4B). Quantification showed that relative to controls, colocalization between these two markers decreased by 19%, 35% and 40%, in TSG101-, ALIX-, and \VPS4B-depleted cells, respectively (Fig 7D). Collectively, these results indicate that a portion of VACV IEVs formed in macrophages occurs by ESCRT-dependent budding into MVBs, strongly suggesting that the MVB-IEV pathway is a functional route to EEV formation in multiple cell types.

## Discussion

Our dual fluorescence–based VACV screen revealed a multitude of cellular functions important for VACV spread. These included SNAP-SNAREs, clathrin- and actin-related proteins, Rabs, autophagy factors, and ESCRT machinery. Consistent with previous reports the importance of actin-branching, clathrin-mediated endocytosis, Rab proteins, retrograde transport, and endosome-to-golgi transport in VACV spread were highlighted (Chen et al, 2009; Sivan et al, 2013; Beard et al, 2014; Leite & Way, 2015; Harrison et al, 2016; Sivan et al, 2016). Additionally, we identified the ESCRT proteins TSG101 and the AAA ATPase, VPS4A.

Cellular ESCRT machinery is involved in membrane shaping and remodeling reactions in mammalian cells whereby it drives "reverse topology" fission, or closure, of double membranes away from cytoplasm (McCullough et al, 2013; Scourfield & Martin-Serrano, 2017; Vietri et al, 2020). The hijacking of ESCRT machinery for viral budding, first reported for HIV-1 and Ebola (Garrus et al, 2001; Martin-Serrano et al, 2001; VerPlank et al, 2001; Demirov et al, 2002), is now considered a common mechanism used by various enveloped and non-enveloped viruses to mediate the release of infectious virus particles from host cells (Votteler & Sundquist, 2013; Weissenhorn et al, 2013; Scourfield & Martin-Serrano, 2017).

For enveloped viruses, ESCRT-mediated budding drives the formation and release of nascent virions at the plasma membrane (Votteler & Sundquist, 2013; Scourfield & Martin-Serrano, 2017). One known exception is HSV-1, which uses ESCRT machinery during multiple stages of egress including primary envelopment at the inner nuclear membrane and secondary envelopment within the

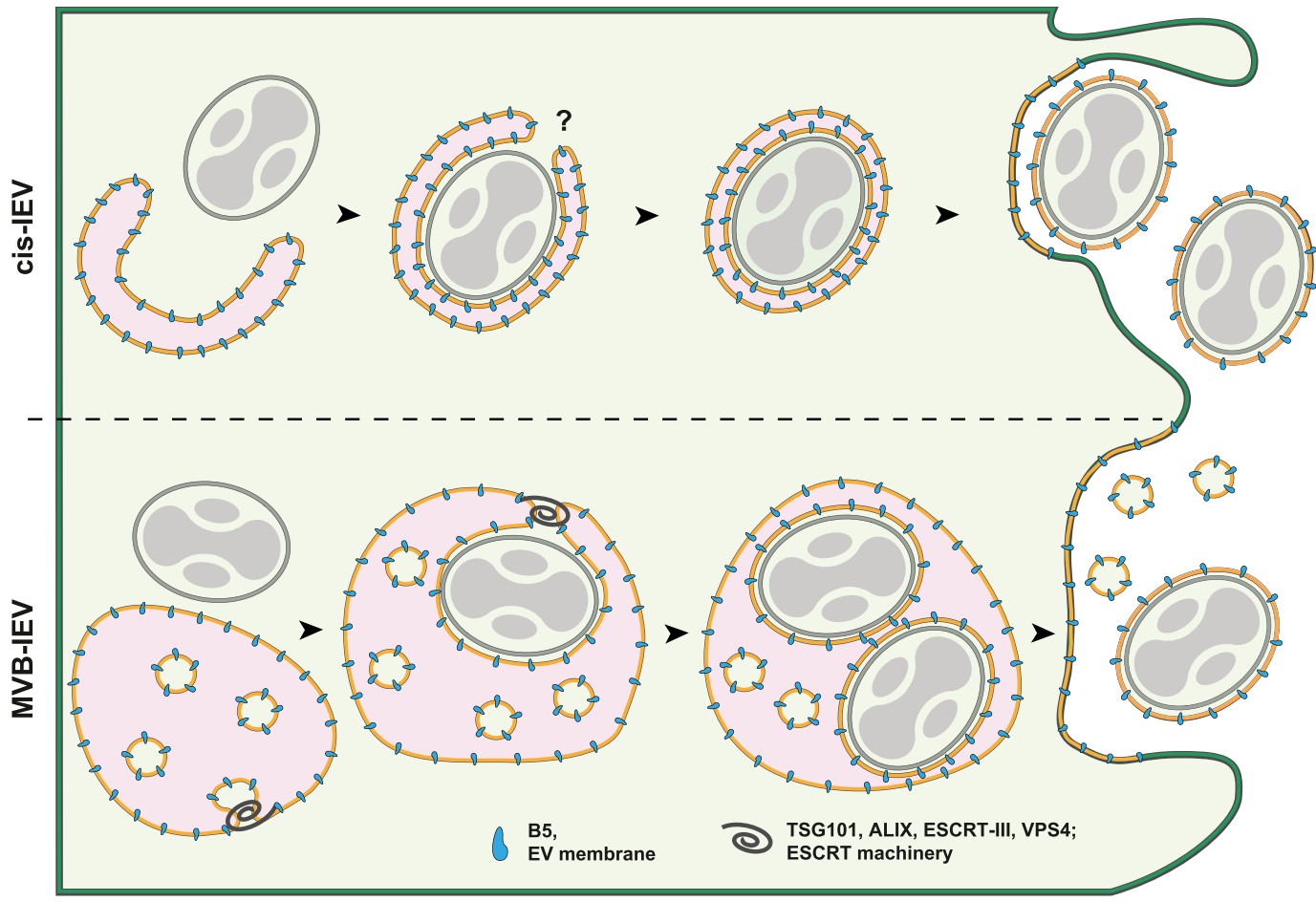

**Figure 8. Model of VACV intracellular enveloped virion (IEV) formation and egress.**
The formation of VACV IEVs proceeds through one of two pathways: Cisternae (CIS)-based wrapping (top), or as described in this report multivesicular body (MVB)–based wrapping (bottom). In both cases, single-membrane intracellular mature virions (IMVs) bud into virus-modified cellular membranes. For CIS wrapping, IMV is enveloped in a tight-fitting, double-membrane cisternae derived from the TGN or early endosome resulting in the formation of the triple-membrane IEV. During MVB-based wrapping, IMV(s) bud into the lumen of the MVB resulting in the acquisition of a tight second membrane, with the limiting membrane of the MVB effectively becoming the third IEV membrane. Although the mechanism and cellular factors that regulate closure of CIS-IEVs is unknown, we show that the formation—and presumably closure—of MVB-IEVs depends on cellular endosomal sorting complexes required for transport machinery. Upon formation, both CIS- and MVB-IEVs transit to the plasma membrane where they undergo fusion leaving behind the outermost membrane to become double-membrane extracellular enveloped virions. We found that MVB-based wrapping accounts for half of all VACV wrapping events and subsequent extracellular enveloped virion formation.

cytoplasm (Calistri et al, 2007; Crump et al, 2007; Pawliczek & Crump, 2009; Crump, 2018; Barnes & Wilson, 2020). Similar to HSV-1, VACV wrapping and egress is a complicated multistep process (see Fig 7). First, single-membrane IMVs are wrapped by a virus-modified double membrane within the cytoplasm. These triple-membrane IEVs then move to the plasma membrane and undergo fusion, leaving behind the outmost membrane resulting in the release of a double-membrane EEV (Smith et al, 2002; Condit et al, 2006; Moss, 2007; Roberts & Smith, 2008; Bidgood & Mercer, 2015).

Consistent with a role for ESCRT in VACV wrapping and/or egress Honeychurch et al (2007) reported that Alix and TSG101 were involved in VACV EEV release. However, compared to HIV-1, for example, VACV does not acquire, but rather leaves behind a membrane at the cell–surface upon exiting. These results suggested to us that ESCRT machinery must be required for a cytoplasmic stage of VACV egress.

Although targeting and assembly factors vary between different ESCRT-dependent processes, ESCRT-III and Vps4 are universally required (Hurley, 2015; Christ et al, 2017). Although our screen suggested that VPS4A was required for VACV spread, as VPS4B was not targeted in the screen, follow-up individual depletion experiments were performed which indicated that VPS4B was rather the critical factor. These experiments revealed that, in HeLa cells, the relative abundance of VPS4A and VPS4B is markedly different, and that a complex compensatory relationship may exist between these paralogs—consistent with findings in yeast and cancer cells (Scheuring et al, 2001; Szymańska et al, 2020).

Consistent with our hypothesis, depletion of ESCRT-III components did not impact IMV formation but resulted in a diminution in VACV EEV production. Strikingly, we found that when CHMP1A, CHMP3, CHMP4C, or CHMP6 were depleted virions began to accumulate on the limiting membrane of enlarged cytoplasmic

structures positive for the VACV EV protein B5, which is required for wrapping (Engelstad & Smith, 1993). Of the three mammalian CHMP4 paralogs (CHMP4A/B/C), CHMP4B is the main filament forming protein involved in ILV formation and HIV-1 budding, whereas CHMP4C has been linked to abscission (Carlton et al, 2008; Morita et al, 2011; Scourfield & Martin-Serrano, 2017; Wenzel et al, 2018). Interestingly, our data indicate that CHMP4C is required for, and CHMP4B represses VACV EEV formation. Studies addressing how these proteins participate in different stages of VACV wrapping, PM budding or release of EEVs may provide additional insight into the divergent functions or regulation of these paralogs.

As VACV replicates in the cytoplasm it is perhaps no surprise that it takes advantage of MVBs. As evidenced by our EM images, VACV wrapping sites are fully "open" to cellular membrane structures and ESCRTs are soluble cytoplasmic proteins that facilitate invagination and wrapping of cargoes. Yet, how VACV is targeted the MVBs and how ESCRT is targeted to VACV remains an open question. Whereas we found B5 in MVBs, a second viral protein required for wrapping, F13, contains a late domain, which in retroviruses serves for the recruitment of ESCRT to viral budding sites (Strack et al, 2003; Honeychurch et al, 2007; Zhai et al, 2008). Determining how these proteins drive MVB-IEV formation will be of future interest.

That we found B5 in ILVs without internalised virions suggests that this protein may have an intrinsic capability to be sorted into or drive ILV budding. Interestingly, it's been reported that inhibition of clathrin-mediated endocytosis results in retention of EEV membrane proteins at the plasma membrane, a 50% drop in EEV formation and a reduction in virus spread (Husain & Moss, 2005). Although it was suggested that these proteins are recycled for use in cisternal wrapping, in light of our identification of MVB-based wrapping, it seems more likely that recycled EEV membrane proteins continue along the endocytic route and contribute to MVB-IEV formation.

As illustrated in Fig 8, we have identified ESCRT-dependent IMV wrapping as the second major form of IEV production and B5-positive MVBs as a novel cellular source of VACV wrapping membrane. As the formation of IEVs and subsequently CEVs is critical for virus spread (Roberts & Smith, 2008; Leite & Way, 2015; Beerli et al, 2019), it is not surprising that VACV has built in redundancy. By using divergent cellular membrane sources and different wrapping mechanisms—TGN/early endosomes versus MVBs—the virus assures sufficient wrapping membrane and secures its ability to infect neighboring cells.

# Materials and Methods

### Cells and viruses

HeLa (CCL-2; American Type Culture Collection [ATCC]) and African green monkey kidney BSC40 (from P. Traktman, Medical University of South Carolina, Charleston, SC, USA) cells were maintained in DMEM supplemented with 10% FBS, 2 mM GlutaMAX and penicillin–streptomycin at 37°C and 5% $CO_2$. BSC40 medium was supplemented with 100 $\mu$M nonessential amino acids and 1 mM sodium pyruvate (Thermo Fisher Scientific). THP-1 monocytes (TIB-202;

ATCC) were maintained in suspension culture in RPMI + GlutaMAX and 10% FBS. HeLa cells have been authenticated by the ATCC and BSC40 cells have not been authenticated. All cell lines were tested regularly and remained mycoplasma free throughout this study. Recombinant VACV strains were based on the VACV strain WR. Recombinant VACVs were generated using homologous recombination as previously described (Mercer & Helenius, 2008; Schmidt et al, 2011). All viruses were produced in BSC40 cells, and mature virions were purified from cytoplasmic lysates through sedimentation as previously described (Mercer & Helenius, 2008).

### Antibodies

Following antibodies were used: anti-GFP rabbit, actin, and Tsg101 (Sigma-Aldrich); Alix and CHMP3 (Santa Cruz Biotechnology); and monoclonal and polyclonal anti-B5 (VMC-20 and R182, respectively) were kind gift of GH Cohen (University of Pennsylvania, Philadelphia, PA, USA). CHMP1A, CHMP2B, CD63, GM130, and TGN46 (Abcam); CHMP1B (ProteinTech Group); EEA1 and Lamp1 (Cell Signaling). Anti-L1 mouse monoclonal antibody (clone 7D11) was purified from a hybridoma cell line that was provided by B. Moss (National Institutes of Health) with permission of A. Schmaljohn (University of Maryland). IRDye-coupled secondary antibodies and Alexa-secondary antibodies were purchased from Invitrogen/Thermo Fisher Scientific.

### Short interfering RNA silencing

HeLa cells were reversed transfected with scrambled (SCR) or various ESCRT short interfering RNA (siRNA) at final concentrations of 40 or 10 nM (Vps4A/B in Fig 2). Transfections were carried out by using Lipofectamine RNAiMAX transfection reagent (Thermo Fisher Scientific). 72-h (or 24 + 24 h for Vps4/B in Fig 2 and CHMP4A/B/C in Figs 3 and 4) post-transfection, cells were either collected for silencing validation or infected with VACV. ON-TARGETplus SMART-pool siRNA for human were purchased from Horizon Discovery (See Tables S1 and S2). AllStars negative control (SCR) and AllStars Death control siRNAs were purchased from QIAGEN. For THP-1s, 187.5 nM SCR or various ESCRT siRNAs were reverse transfected using Viromer Blue (# VB-01LB; Lipocalyx). Briefly, THP-1 monocytes in suspension were collected by centrifugation, resuspended in fresh culture media, and seeded into wells containing Viromax/siRNA mixture. Following 72 h incubation, THP-1 monocytes were differentiated into macrophages by the addition of phorbol 12-myristate 13-acetate (PMA) in RPMI/50% FBS at 75 ng/ml final concentration. Cells were briefly centrifuged and incubated at RT for 30 m to promote even seeding and attachment. The medium was exchanged at 24 h after transfection and cells incubated for an additional 24 h before infection with VACV.

### VACV infections

HeLa cells were seeded on 96-well plates, coverslips, or six-well plates depending on the assay. Cells were infected with VACV WR or recombinant viruses in DMEM without supplements. After 1 h of infection, the medium was changed to full DMEM. At indicated time points, cells were fixed with methanol-free formaldehyde for 20 min. All VACV immunofluorescence infections were performed at

MOI 10 in both HeLa and THP-1 monocyte-derived macrophages. Spread screen samples were infected with VACV E-EGFP L-mCherry at MOI 0.02 and the infection medium was changed to medium containing AraC (10 μM; Sigma-Aldrich) at 8 hpi.

## Immunofluorescence labelling and imaging

Fixed cells were permeabilized with 0.1% Triton in PBS for 10 min before staining. Both primary and secondary antibodies were diluted in 3% BSA in PBS. Antibodies were incubated for 1 h at RT and washed three times with 0.5% BSA/PBS after incubations. Samples were mounted with mounting media containing DAPI (Invitrogen). Spread screen samples were labelled with Hoechst (Invitrogen). Confocal fluorescence microscopy was performed using a 100× oil immersion objective (NA 1.45) on a VT-iSIM microscope (Nikon Eclipse TI; Visitech), using 405-, 488-, 561-, and 647-nm laser frequencies for excitation.

## High-content image acquisition (spread screen)

Confocal fluorescence microscopic images were acquired using an Opera Phenix (PerkinElmer) high-content screening system using Harmony 4.9 software. 96-well Viewplate (PerkinElmer) microtitre plate geometries were used as plate type setting and was autofocused by the default two-peak method. Standard 20× (NA = 0.4) air objective was used without camera binning for confocal mode imaging in three channels. DAPI channel images (laser excitation = 375 nm, emission = 435–480 nm) were acquired at 3.0 μm focus height with 100% laser power and 1,200 ms acquisition time. Alexa 488 channel images (laser excitation = 488 nm, emission = 500–550 nm) were acquired at 6.0 μm focus height with 80% laser power and 400 ms acquisition time. The mCherry channel images (laser excitation = 561 nm, emission = 570–630 nm) were acquired at 6.0 μm focus height with 80% laser power and 800 ms acquisition time. Twenty-five fields of view were imaged in a 5 × 5 square layout at the center of each well. The total image acquisition time duration of the 96 wells of one plate was 114 min, with total image data size of 39.1 GB per plate.

## High-content analysis

Images were processed using a SuperServer 4048B-TR4FT system with ImageJ version 1.49 (https://imagej.nih.gov/ij/). A full plate of 864 images was analysed as a set on each ImageJ instance with a custom ImageJ macro. The image set of a plate was read into the memory and converted into a three channel hyperstack. The calculation was iterated on all fields of views as the following. A watershed segmentation algorithm with suitably chosen parameter was run on blue nuclei that identified the location of each nucleus based on the local maximum pixel intensities on channel 1. Then the pixel intensity under each nucleus location was measured on channel 2 (EGFP) and channel 3 (mCherry), resulting in early infection (green, EGFP) and late infection (magenta, mCherry) cell intensity measurements respectively. The resulting cell intensity data were further processed with R (https://www.r-project.org/) version 3.2.3, where the ratios "green cell number/blue cell number" and "magenta cell number/blue cell number" were calculated with a custom R script for each well. A suitably chosen 500 and 600

intensity threshold value was used to specify green cells and magenta cells respectively, that showed adequately strong signal.

## Western blotting

For siRNA depletion validation, samples were scraped into Blue Loading Buffer (Cell Signaling). Lysates were ultrasonicated for 15 min, boiled in reducing sample buffer for 10 min, and centrifugated at 14,000$g$ for 10 min. Samples were then run on 4–12% Bis-Tris gels (Thermo Fisher Scientific) and transferred to nitrocellulose. Membranes were blocked with 5% BSA in 1% Tween/TBS and incubated with primary antibodies either 1 h RT or overnight according to the manufacturer's recommendations. IRDye-coupled secondary antibodies were used for detection on a LI-COR Odyssey imaging system.

## Mature virion/extracellular enveloped virion 24-h yield

Confluent BSC40 cells in six-well plates were infected with virus at MOI 1 and fed with 1 ml full medium. At 24 hpi, the supernatant containing EEVs was collected and cleared of cells by 2× centrifugation at 400$g$ for 10 min. Remaining MVs and partially closed EEVs were neutralized with 7D11 antibody (4 μg/ml) for 1 h +37C. For IMVs, cells were collected by scraping, centrifuged, and resuspended in 100 μl 1 mM Tris (pH 9.0), before 3× freeze–thaw. The plaque forming units/ml (pfu/ml) were determined by crystal violet staining of plaques, 48 hpi after serial dilution on confluent monolayers of BSC40 cells.

## qRT-PCR

Total RNA was extracted from siRNA-treaded cells using the RNeasy Mini kit (QIAGEN), 48, 72 h, or 5 d (THP-1) post-transfection. 1 μg of total RNA was used for cDNA synthesis with SuperScript reverse transcriptase (Thermo Fisher Scientific) and oligo(dT) primer (Invitrogen). qRT-PCR was performed using MESA Blue SYBR Green Mastermix (Eurogentec). Pre-designed primers were purchased from OriGene Technologies. Reactions were analysed upon an ABI 7000 real-time PCR machine (Thermo Fisher Scientific) using the cycle conditions suggested by Eurogentec (MESA Blue). Results were normalized against GAPDH expression. Primer sequences can be found in Table S2.

## ZedMate quantification

Detection and analysis of the individual virus particles was performed using custom developed ImageJ/Fiji plugin ZedMate (https://www.biorxiv.org/content/10.1101/820076v2). In this plugin, particle detection in three-dimensional micrographs is first performed using the reference channel (e.g., mCherry core) in each individual lateral plain using Laplacian of Gaussian spot detection engine. Then the lateral spots are connected in axial direction. Finally, the intensity measurement is performed for all channels for each particle. According to the signal in the specific channel, particles are sorted into their respective types.

## Electron microscopy

HeLa cells were infected with mCh-A5 VACV with a MOI of 10 for 8 h for HeLa cells or a MOI of 20 (BSC40 titer) for 24 h for THP-1 cells.

Samples were fixed with 1.5% glutaraldehyde/2% formaldehyde (EM-grade; TAAB) in 0.1 M sodium cacodylate for 20 min at RT and secondarily fixed for 1 h in 1% osmium tetraoxide/1.5% potassium ferricyanide at 4°C. Samples were then treated with 1% tannic acid in 0.1 m sodium cacodylate for 45 min at room temperature and dehydrated in sequentially increasing concentration of ethanol solutions, and embedded in Epon resin. Epon stubs were polymerized by baking at 60°C overnight. The 70 nm thin sections were cut with a Diatome 45° diamond knife using an ultramicrotome (UC7; Leica). Sections were collected on 1 × 2 mm formvar-coated slot grids and stained with Reynolds lead citrate. For cryo-EM cells were fixed with 4% formaldehyde in phosphate buffer for 2 h at room temperature, infused with 2.3 M sucrose, supported in 12% (wt/vol) gelatin, and frozen in liquid nitrogen. Ultrathin (70 nm) cryosections were cut at −120°C and picked up in 1:1 2% sucrose: 2% methylcellulose. Sections were labelled with primary antibody (mouse anti-B5), followed by rabbit anti-mouse intermediate antibody (DAKO) and protein A gold (University of Utrecht). Finally, sections were contrast stained in 1:9 solution of 4% uranyl acetate: 2% methylcellulose solution pH 4.0. TEM micrographs were obtained using a Tecnai T12 Thermo Fisher Scientific equipped with a charge-coupled device camera (SIS Morada; Olympus).

## Supplementary Information

## Acknowledgements

We thank David Albrecht for his illustration, which was adapted for this manuscript, and Daniel Fisch for the template used to generate the BioRxiv manuscript. We also want to thank GH Cohen and B Moss for the VACV antibodies. M Huttunen and J Mercer are supported by core funding to Medical Research Council (MRC) Laboratory for Molecular Cell Biology at University College London (MC_UU12018/7), A Yakimovich, and J Mercer by the European Research Council (649101-UbiProPox). IJ White and J Kriston-Vizi are supported by MRC core funding to the MRC Laboratory for Molecular Cell Biology at University College London, award code (MC_U12266B). J Martin-Serrano is supported by the Wellcome Trust (WT102871MA). WI Sundquist is supported by National Institutes of Health (NIH) grant R37 AI 51174. The paper is supported by MRC-UCL University Unit grant Ref MC_U12266B and MRC Dementia Platform Grant UK MR/M02492X/1 (High-Content Biology laboratory, University College London). This research was funded in part by The Wellcome Trust. A CC BY license is applied to the author approved manuscript arising from this submission, in accordance with the grant's open access conditions. EM Frickel is supported by a Wellcome Trust Senior Research Fellowship (217202/Z/19/Z). This work was supported by the Francis Crick Institute, which receives its core funding from Cancer Research UK (FC001076 to EM Frickel), the UK Medical Research Council (FC001076 to EM Frickel), and the Wellcome Trust (FC001076 to EM Frickel).

### Author Contributions

M Huttunen: conceptualization, formal analysis, funding acquisition, validation, methodology, and writing—original draft, review, and editing.

J Samolej: formal analysis, validation, investigation, methodology, and writing—review and editing.
RJ Evans.: formal analysis, validation, investigation, and methodology.
A Yakimovich: data curation, formal analysis, investigation, and methodology.
IJ White: data curation, formal analysis, investigation, and methodology.
J Kriston-Vizi: data curation, formal analysis, and methodology.
J Martin-Serrano: conceptualization and resources.
WI Sundquist: conceptualization and methodology.
E-M Frickel: conceptualization, resources, supervision, funding acquisition, methodology, and writing—review and editing.
J Mercer: conceptualization, data curation, formal analysis, supervision, funding acquisition, visualization, project administration, and writing—original draft, review, and editing.

## Conflict of Interest Statement

The authors declare that they have no conflict of interest.

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
