## [Reviewer comments · Life Science Alliance]

Life Science Alliance

Vaccinia Virus Hijacks ESCRT-mediated Multivesicular Body Formation for Virus Egress

Moona Huttunen, Jerzy Samolej, Robert Evans, Artur Yakimovich, Ian White, Janos Kriston-Vizi, Juan Martin-Serrano, Wesley Sundquist, Eva-Maria Frickel, and Jason Mercer

DOI: <https://doi.org/10.26508/lsa.202000910>

Corresponding author(s): Jason Mercer, University of Birmingham and Moona Huttunen, Institute of Microbiology and Infection, University of Birmingham, Birmingham, United Kingdom

Review Timeline:

Submission Date:	2020-09-18
Editorial Decision:	2020-10-02
Revision Received:	2021-04-23
Editorial Decision:	2021-05-27
Revision Received:	2021-06-03
Accepted:	2021-06-04

Transaction Report:

Please note that the manuscript was previously reviewed at another journal and the reports were taken into account in the decision-making process at Life Science Alliance.

Reviewer #1 Review

Comments to the Authors (Required):

Huttunen et al. described their findings that the ESCRT pathway contributes to the non-cisternal egress of the prototypic poxvirus, vaccinia virus (VACV). They report that the non-cisternal egress in HeLa cells contributes to ~20% of IEVs found in the cytosol of infected cells. Following the description in 2007 that other two ESCRT-related factors, ALIX and TSG101, reduced the formation of extracellular enveloped viruses (EEV), the authors now show that the AAA+ ATPase VPS4B is the final factor contributing to the formation of extracellular and intracellular enveloped virions (IEV) but not intracellular mature virions (IMV). In terms of mechanism, the authors found that when ESCRT-III components CHMP1A, CHMP3, CHMP4C or CHMP6 were depleted, virions began to accumulate on membranes of enlarged cytoplasmic structures positive for the VACV EV protein

B5. The study is well performed, very well written and the techniques used are suitable to answer the question. It is well appreciated the amount of work required to quantify virions inside MVB using electron microscopy. However, at present the study is more descriptive than mechanistic being an increment to the list of different viruses that use some factors of the ESCRT machinery.

MAJOR POINTS

The authors have described a mechanism contributing to 20% of EEVs from VACV that is independent of cisternal-wrapping and relies on the ESCRT machinery. Two points are noted. The reduction in EEVs by depleting VPS4B and the observation of reduced numbers of virions inside MVB per se is a correlation that may not reflect causation. For example, VPS4A depletion leads to reduced number of MVB containing virions and is not translated in a reduction of EEVs. The authors could provide more evidence to substantiate their claims by addressing how IEVs-VACV are targeted to multivesicular bodies (MVBs). By doing so, the authors could formally demonstrate, by two independent ways that mutating late domains in viral proteins/targeting host factors by siRNA would lead to reduction in EEVs and MVB containing virions. The second point is that authors have investigated their findings in HeLa cells, and it would be interesting to have their findings validated in other cell types, including for example macrophages, that are important for the spread of poxviruses. The points this reviewer raises are the listed below:

1. How does depleting both VPS4A and VPS4B impact EEV production?
2. Is the EEV decrease upon VPS4B observed in other cell types? Macrophages or cells from the airway epithelia?
3. In the J Virol. paper of 1994 paper by SCHMELZ et al, the authors noted virions inside endosomal membranes, which is consistent with the observations in the present manuscript. One factor that the authors may now answer because they have the tools, is whether usage of endosomal compartments may reflect viral entry into new cells or egress.
4. Using mutant viruses would validate the findings and show how IEVs are targeted to MVBs.

MINOR POINTS:

Discussion:

1. Please change VACV is targeted the MVBs to VACV is targeted to MVBs in the sentence "Yet, how VACV is targeted the MVBs and how ESCRT is targeted to VACV remains an open question"

Reviewer #2 Review

Comments to the Authors (Required):

The manuscript by Huttunen et al. details the authors investigation into cellular factors required for vaccinia virus (VACV) spread in HeLa cells. VACV has a complicated morphogenesis pathway involving several cellular membrane compartments. This includes the formation of single membrane bound virions termed IMVs in virus factories that are infectious and released by cell lysis.

Additionally a proportion of IMVs undergo wrapping/budding at intracellular membranes to form a double membraned virion within the lumen of vesicles/vacuoles (triple membrane structures termed IEVs). IEVs fuse with the plasma membrane to secrete the double membraned virions from cells, which are termed EEVs once released into culture media. The mechanisms of IEV formation and EEV secretion and spread to uninfected cells are relatively poorly defined.

The authors conducted an siRNA screen for host factors required for VACV spread, identifying ESCRT proteins as hits, which formed the focus of this study. They provide evidence suggesting ESCRT function is required for the wrapping/budding of a subset of IEVs, with inhibition of this

process causing reduced secretion of EEVs and reduced spread to neighboring cells. Using electron microscopy the authors then found that an approximately equal number of IEVs were present in multivesicular bodies (MVBs) as were present in "cisternal-wrapped" IEVs - the more classically defined triple membrane structures observed in previous EM studies of VACV morphogenesis. The authors further show that ESCRT function is required for the MVB wrapping but not cisternal wrapping.

Overall the data is of high quality and will be of interest to colleagues studying the assembly pathways of poxviruses and other nucleocytoplasmic large DNA viruses. However, the importance of the observed MVB wrapping pathway for poxvirus biology would benefit from further investigation. For example, are MVB-IEVs commonly observed in other cell types (in other cell culture models or infected tissue samples) or is this specific to HeLa cells? Is there evidence that MVB-IEVs fuse with the plasma membrane to secrete virions? Does the composition of the EEV originating from MVB or cisternal wrapping differ e.g. can EEVs be classified by late endosomal or TGN/early endosome protein or lipid markers? Does the secretion of EEVs correlate with increased secretion of other MVB contents like intraluminal vesicles? While these are likely difficult questions to answer, further data in these areas would help broaden the appeal of this work for the cell biology community.

Other comments;

1. The way the siRNA screen is designed suggests it would identify host factors involved in mechanisms that are required specifically for the entry of CEV/EEV forms of VACV but not MVs. This would be worth highlighting.
2. The siRNA screen identified NEDD4 whereas the validation was performed with NEDD4L siRNA - is there a reason for this?
3. Why was CHMP2A not included in Figure 3?
4. Quantification of F13 co-localization with cellular markers and analysis of an additional IEV specific virus protein (e.g. B5) would enhance the data in Figure 5A and enable comparison with immune-EM data on B5 localization in Figure 6C.
5. Can fusion of IEVs with the plasma membrane be observed in the EM samples? If so what proportion are MVB-IEVs compared to cis-IEVs?
6. The quantification of cis-IEVs should be included in Figure 6E.
7. The effects of VPS4A knockdown shown in Figures 1C, 2C and 2D don't seem to correlate with the data in Figure 6E. This needs an explanation or comment.

Reviewer #3 Review

Comments to the Authors (Required):

Poxviruses undergo a complicated and not well-characterized membrane wrapping process as part of their maturation. This study began with a screen to identify cellular factors involved in virus egress and ultimately focused on the role of the ESCRTs in EEV formation and wrapping of IMVs at MVBs. The identification of MVBs as a potential source of poxvirus membrane as an alternative to the traditional cisterna wrapping model would be a significant finding and the data are compelling, however there are several areas that need to be addressed to solidify the interpretations and support the conclusions of the paper.

1. A major concern is the integration of the siRNA screen with the rest of the manuscript. The ESCRT components identified in the screen to affect virus egress were Tsg101, Vps4A and

NEDD4. Tsg101 had already been shown to affect production of EEVs. However, no additional work was done on NEDD4 and it was VPS4B and not VPS4A that was later shown to be most important for EEV formation. It does not appear that VPS4B or the ESCRT-III components were included in the screen. If they were included, there is no discussion as to why they were not hits. If they were not included, then the significance of the screen is significantly weakened with respect to rest of the manuscript. In fact, it appears that Figures 1C and 2A are in direct conflict to each other. Is this a result of the difference in the time point (48 vs 72)? If so, please discuss. Based on Figure 2A, siVPS4A would not have been a hit in the initial screen as it did not result in a 25% reduction in spread. Thus, as currently presented, it is not clear if the screen in Figure 1 is necessary and supportive of the rest of the manuscript.

2. On a related note, if VPS4B is increased 2.5-fold upon VPS4A depletion and is the more important of the two factors, why is any phenotype observed at all with VPS4A depletion in Figure 1C? Please discuss why the increase in VPS4B would not compensate.

3. The lack of colocalization between the core and IEV wrapping machinery in the VPS4B sample of Figure 2D led to the conclusion that "VPS4B is a pro-viral factor involved in VACV EEV formation" However, the nucleoplasm of the VPS4B knockdown in Figure 2D appears dispersed, particularly in comparison to knockdown with VPS4A. How is this related to EEV formation? Could an alternative interpretation be that VPS4B disrupts another viral process upstream of wrapping, resulting in dispersed nucleoplasm and the lack of colocalization? Knockdown of VPS4 affects many aspects of normal cell biology and additional data is needed to support the claim that VPS4B is specifically involved in EEV formation.

4. The divergent function of CHMP4B is quite striking. Given that CHMP4B is the main filament forming protein in ILV formation, which is analogous to the mechanism you are proposing for wrapping into MVBs, why would CHMP4B repress? What is known about the isoform distribution of CHMP4 isoforms in the HeLa cells used in Figure 3? Are 4A and 4C more abundant? This interesting but counterintuitive observation would be strengthened if repeated in CHMP4 knockdown during infection of other cell types used for poxvirus infection. Along this note, what happens to levels of the CHMP proteins during infection? Specifically, does the virus downregulate the repressive CHMP4B or upregulate CHMP4A or C? What about the other CHMPs?

5. Please add scale bars to the insets in Figure 5.

6. In Figure 6, what happens to the total number of MVBs upon knockdown of VPS4, specifically VPS4B? While zero IEV-MVBs were reported after VPS4B knockdown (and only 4% after VPS4A knockdown), one would expect that you also would have significant reduction in MVBs since ILV formation would also be disrupted upon depleting VPS4. Thus, reporting percentage of IEV-MVBs is not informative without accounting for a difference (if any) in total MVBs.

7. Importantly, what are the fate of the MVBs containing IEVs? The contents of many MVBs are destined for lysosomal degradation while others are released as extracellular content. Does the virus alter the fate of MVB containing IEVs so that the IEVs are predominantly released as EEVs? Or is a significant portion of MVB-associated IEVs degraded in lysosomes? Perhaps this is why CHMP4B represses EEV formation. While it is clear from the data that EEVs are reduced upon knockdown of ESCRT-III and VPS4, an alternative interpretation consistent with the data in the manuscript is that MVB-associated IEVs are actually a dead end destined for degradation and that the reduction in EEVs observed with VPS4B and CHMP knockdowns are an indirect effect of disrupting another viral or cellular process. More work is required to show that MVB-associated IEVs

are released as EEVs.

October 2, 2020

Re: Life Science Alliance manuscript #LSA-2020-00910-T

Prof. Jason Mercer
University of Birmingham
Institute of Microbiology and Infection
Edgbaston
Birmingham B15 2TT
United Kingdom

Dear Dr. Mercer,

Thank you for transferring your manuscript entitled "Vaccinia Virus Hijacks ESCRT-mediated Multivesicular Body Formation for Virus Egress" to Life Science Alliance (LSA).

For a brief overview, this manuscript was reviewed at a partner journal, where the reviewers thought that the study lacked the mechanistic exploration of the effects seen. The authors then transferred the manuscript along with the referee reports from the previous journal to LSA.

Given the new data showing the role of ESCRT components in MVB formation for vaccinia virus egress, we think the study should be publishable in LSA pending following revisions:

- + We agree with the concern brought up by both reviewers 1 and 2 about whether these data pertain only to HeLa cells or if they would also hold true for other cell types, i.e. macrophages, which are important for spread of pox viruses in vivo (Rev 1 pt 2, Rev 2, general point). We encourage you to repeat some of your salient data in macrophages, in vitro data should be sufficient
- + Figuring out whether usage of endosomal components reflects viral entry in new cells or egress (R1 pt 3), how IEVs-VACV are targeted to MVBs (R1 pt 4), whether MVBs fuse with plasma membrane to secrete virions (R2 general comments, and pt 5), difference in compositions of EEV originating from MVB or cisternal wrapping, whether secretion of EEVs correlate with increased secretion of other MVB contents (R2 general points), and determining the fate of MVBs containing IEVs (R3 pt 7) would not be required for publication in LSA, unless any of these data is readily available - however, a discussion on these points in the revised manuscript is encouraged.
- + We encourage you to elaborate and provide a discussion on the concerns raised by R3 in pts 3, 4; and provide clarifications for the questions raised by R3 in pt 2, 6
- + Both reviewers 2 and 3 have raised a concern about the mis-match between the data found in the screen and the rest of the study (R2 pt 2, 3 and R3 pt 1) - we would appreciate if you can provide a discussion on this disparity
- + Please provide quantifications requested by R2 in pts 4 and 6
- + Please also provide a detailed point-by-point rebuttal of the reviewers comments

We understand that these data might need to be re-reviewed, in which case, we will, of course, walk the reviewer(s) through our transfer process.

We would be happy to discuss the individual revision points further with you should this be helpful. While you are revising your manuscript, please also attend to the below editorial points to help expedite the publication of your manuscript. Please direct any editorial questions to the journal office. The typical timeframe for revisions is three months. Please note that papers are generally considered through only one revision cycle, so strong support from the referees on the revised version is needed for acceptance.

Please reach out to the LSA editorial office if you have any additional questions.

Thank you for considering Life Science Alliance as an appropriate venue for your research. We look forward to receiving your revised manuscript.

Sincerely,

Shachi Bhatt, Ph.D.
Executive Editor
Life Science Alliance
<https://www.life-science-alliance.org/>
Tweet @SciBhatt @LSAJournal

- A letter addressing the reviewers' comments point by point.
- An editable version of the final text (.DOC or .DOCX) is needed for copyediting (no PDFs).
- High-resolution figure, supplementary figure and video files uploaded as individual files: See our detailed guidelines for preparing your production-ready images, <https://www.life-science-alliance.org/authors>
- Summary blurb (enter in submission system): A short text summarizing in a single sentence the study (max. 200 characters including spaces). This text is used in conjunction with the titles of papers, hence should be informative and complementary to the title and running title. It should describe the context and significance of the findings for a general readership; it should be written in the present tense and refer to the work in the third person. Author names should not be mentioned.

B. MANUSCRIPT ORGANIZATION AND FORMATTING:

We thank the editor and reviewers for their helpful comments and suggestions. In line with the suggestions, we have added additional discussion and new experiments to the manuscript which have served to strengthen our original conclusion, that VACV subjugates ESCRT-mediated MVB formation for IEV formation. Please find a point-by-point response below.

Reviewer #1 (Comments to the Authors (Required)):

Huttunen et al. described their findings that the ESCRT pathway contributes to the non-cisternal egress of the prototypic poxvirus, vaccinia virus (VACV). They report that the non-cisternal egress in HeLa cells contributes to ~20% of IEVs found in the cytosol of infected cells. Following the description in 2007 that other two ESCRT-related factors, ALIX and TSG101, reduced the formation of extracellular enveloped viruses (EEV), the authors now show that the AAA+ ATPase VPS4B is the final factor contributing to the formation of extracellular and intracellular enveloped virions (IEV) but not intracellular mature virions (IMV). In terms of mechanism, the authors found that when ESCRT-III components CHMP1A, CHMP3, CHMP4C or CHMP6 were depleted, virions began to accumulate on membranes of enlarged cytoplasmic structures positive for the VACV EV protein B5. The study is well performed, very well written and the techniques used are suitable to answer the question. It is well appreciated the amount of work required to quantify virions inside MVB using electron microscopy. However, at present the study is more descriptive than mechanistic being an increment to the list of different viruses that use some factors of the ESCRT machinery.

MAJOR POINTS

The authors have described a mechanism contributing to 20% of EEVs from VACV that is independent of cisternal-wrapping and relies on the ESCRT machinery. Two points are noted. The reduction in EEVs by depleting VPS4B and the observation of reduced numbers of virions inside MVB per se is a correlation that may not reflect causation. For example, VPS4A depletion leads to reduced number of MVB containing virions and is not translated in a reduction of EEVs. The authors could provide more evidence to substantiate their claims by addressing how IEVs-VACV are targeted to multivesicular bodies (MVBs). By doing so, the authors could formally demonstrate, by two independent ways that mutating late domains in viral proteins/targeting host factors by siRNA would lead to reduction in EEVs and MVB containing virions.

We thank the reviewer for this suggestion. We did not pursue this line of experimentation given that Honeychurch *et al.* 2007 first reported that Alix and TSG101 were required for VACV EEV formation. As discussed in paragraphs 4 of our discussion, they also demonstrated that a late domain within F13 (a VACV protein required for wrapping) gave the same phenotype as ALIX and TSG101 depletion (reduced number of EEVs) and concluded that ESCRT was required for EEV release from the PM.

However, as stated in the discussion in reference to this previous manuscript, "VACV does not acquire, but rather leaves behind a membrane at the cell-surface upon exiting. These results indicated that the ESCRT machinery must be required for a cytoplasmic stage of VACV egress, (i.e IEV formation)". It therefore reasons, as we discuss in paragraph 7 of the discussion that F13 late-domain mediated wrapping is occurring at MVBs.

The second point is that authors have investigated their findings in HeLa cells, and it would be interesting to have their findings validated in other cell types, including for

example macrophages, that are important for the spread of poxviruses.

We thank the reviewer for this suggestion. We have now investigated MVB-IEV formation in monocyte-derived macrophages (MDMs). We show by TEM that both Cis- and MVB-IEVs are formed in MDMs. Using core/EV membrane colocalization as a proxy for IEV formation, akin to Figures 1 and 4, we also show that depletion of ESCRT components in MDMs results in reduced IEV formation. This data is consistent with our findings in HeLa cells and serve to strengthen the conclusion of the manuscript. This data is now included as Figure 7 in the revised manuscript.

The points this reviewer raises are the listed below:

1. How does depleting both VPS4A and VPS4B impact EEV production?

We performed this experiment but found that cells did not tolerate double VPS4A/B depletion preventing us from pursuing these experiments.

2. Is the EEV decrease upon VPS4B observed in other cell types? Macrophages or cells from the airway epithelia?

We now demonstrate that IEV formation is reduced in monocyte-derived macrophages (MDMs) upon VPS4B depletion. Unfortunately, siRNA knockdown combined with 24h VACV productive yield experiments in MDMs were highly variable across different siRNAs. As we could not conclude anything from this data, we moved to single-cell IEV formation as outlined in the response to point 2.

3. In the J Virol. paper of 1994 paper by SCHMELZ et al, the authors noted virions inside endosomal membranes, which is consistent with the observations in the present manuscript. One factor that the authors may now answer because they have the tools, is whether usage of endosomal compartments may reflect viral entry into new cells or egress.

We suspect the reviewer is referring to the 1993 Tooze paper which suggest that VACV wrapping membranes are derived from early endosomes. The Schmelz paper argues that wrapping membranes are derived from the TGN and that the observations of Tooze were due to the use of non-specific fluid phase markers. That said the laboratory of B. Moss has demonstrated that EEV markers can be recycled from the PM to early endosomes where wrapping can occur during later stages of infection.

Our lab has extensively studied the endocytic entry of both VACV IMVs and EEVs (Mercer and Helenius, Science 2008; Mercer *et al.* PNAS 2010; Schmidt *et al.* EMBO 2011; Rizopoulos *et al.* Traffic 2015). Together with the data herein, it appears that VACV subjugates cellular endocytic compartments for internalization and both exo- and endocytic compartments for egress.

4. Using mutant viruses would validate the findings and show how IEVs are targeted to MVBs.

Please see the response to point 1 above. Experiments involving mutation of the only late domain-containing VACV wrapping protein resulted in decreased EEV yield, consistent with our findings.

MINOR POINTS:

Discussion:

1. Please change VACV is targeted the MVBs to VACV is targeted to MVBs in the sentence "Yet, how VACV is targeted the MVBs and how ESCRT is targeted to VACV remains an open question"

Thank you for pointing this out. It has been corrected

Reviewer #2 (Comments to the Authors (Required)):

The manuscript by Huttunen et al. details the authors investigation into cellular factors required for vaccinia virus (VACV) spread in HeLa cells. VACV has a complicated morphogenesis pathway involving several cellular membrane compartments. This includes the formation of single membrane bound virions termed IMVs in virus factories that are infectious and released by cell lysis. Additionally a proportion of IMVs undergo wrapping/budding at intracellular membranes to form a double membraned virion within the lumen of vesicles/vacuoles (triple membrane structures termed IEVs). IEVs fuse with the plasma membrane to secrete the double membraned virions from cells, which are termed EEVs once released into culture media. The mechanisms of IEV formation and EEV secretion and spread to uninfected cells are relatively poorly defined.

The authors conducted an siRNA screen for host factors required for VACV spread, identifying ESCRT proteins as hits, which formed the focus of this study. They provide evidence suggesting ESCRT function is required for the wrapping/budding of a subset of IEVs, with inhibition of this process causing reduced secretion of EEVs and reduced spread to neighboring cells. Using electron microscopy the authors then found that an approximately equal number of IEVs were present in multivesicular bodies (MVBs) as were present in "cisternal-wrapped" IEVs - the more classically defined triple membrane structures observed in previous EM studies of VACV morphogenesis. The authors further show that ESCRT function is required for the MVB wrapping but not cisternal wrapping.

Overall the data is of high quality and will be of interest to colleagues studying the assembly pathways of poxviruses and other nucleocytoplasmic large DNA viruses. However, the importance of the observed MVB wrapping pathway for poxvirus biology would benefit from further investigation. For example, are MVB-IEVs commonly observed in other cell types (in other cell culture models or infected tissue samples) or is this specific to HeLa cells?

This was also raised by reviewer 1. We have now investigated MVB-IEV formation in monocyte-derived macrophages (MDMs). We show by TEM that both Cis- and MVB-IEVs are formed in MDMs. Using core/EV membrane colocalization as a proxy for IEV formation, akin to Figures 1 and 4, we also show that depletion of ESCRT components in MDMs results in reduced IEV formation. This data is consistent with our findings in HeLa cells and serve to strengthen the conclusion of the manuscript. This data is now included as Figure 7 in the revised manuscript.

Is there evidence that MVB-IEVs fuse with the plasma membrane to secrete virions?

Direct observation of MVB-IEV fusion and EEV release would be very difficult to address experimentally for reasons that are also relevant to the question below. I don't know of any existing images of VACV fusion with the PM during egress (Cis or MVB). I suspect this has to do with the speed of the fusion reaction and the fact that the only way to observe this is in a fixed state by EM. You either see intercellular EVs

or membrane bound extracellular EVs. In addition (and related to the point below), when MVB-IEVs fuse with the PM, they would be – as far as we can ascertain - indistinguishable from cisternal derived EEVs.

That said, we do show several ESCRT components are required for MVB-IEV formation and that depletion of these proteins in HeLa cells results in decreased EEV yields (Figures 1-4). We believe this provides, although indirect, rather convincing evidence that MVB-IEVs are ultimately released as EEVs.

Does the composition of the EEV originating from MVB or cisternal wrapping differ e.g. can EEVs be classified by late endosomal or TGN/early endosome protein or lipid markers?

The simple answer here is, we don't know. Based on the result below, and previous work by the group of G. Smith which indicates that cellular proteins are excluded from Cis-EEVs (Krauss et al. J Gen Virol., 2002) we would suspect that the protein composition of Cis- and MVB-derived EEVs would be identical (i.e viral proteins only with cell markers excluded). It is conceivable that the lipid composition would differ based on the different origins of the wrapping membrane. However, as there is currently no way to separate the two populations (protein composition, morphology, etc) its not possible to do this currently.

Does the secretion of EEVs correlate with increased secretion of other MVB contents like intraluminal vesicles?

[Figure removed by editorial staff per authors' request]

While these are likely difficult questions to answer, further data in these areas would help broaden the appeal of this work for the cell biology community.

Other comments;

1. The way the siRNA screen is designed suggests it would identify host factors involved in mechanisms that are required specifically for the entry of CEV/EEV forms of VACV but not MVs. This would be worth highlighting.

We have added this sentence to the introduction to highlight this:

“Quantification of the number of primary (magenta) and secondary infected (green) cells allowed us to differentiate between defects in primary infection by IMVs and defects in virus spread which could be caused by attenuated virion formation or entry of CEVs or EEVs into surround cells.”

2. The siRNA screen identified NEDD4 whereas the validation was performed with NEDD4L siRNA - is there a reason for this?

We thank the reviewer for pointing this out. This was a clerical error. The validation was performed with siRNA directed against NEDD4. We have edited the figure to reflect this.

3. Why was CHMP2A not included in Figure 3?

We attempted siRNA depletion of CHIMP2 using multiple conditions. Cell death associated with its loss, coupled with VACV infection wasn't possible.

4. Quantification of F13 co-localization with cellular markers and analysis of an additional IEV specific virus protein (e.g. B5) would enhance the data in Figure 5A and enable comparison with immune-EM data on B5 localization in Figure 6C.

As suggested, we have now quantified the co-localization of F13 with cellular markers used in 5A. Consistent with cisternae and MVB-based wrapping colocalization with TGN46 (trans-golgi marker) and CD63 (MVB marker) were found to be most abundant. This data is presented as Fig. 5B in the revised manuscript.

5. Can fusion of IEVs with the plasma membrane be observed in the EM samples?

As addressed above, to our knowledge VACV IEV-PM fusion has not been visualized, likely due to the speed of the process and the need to use static EM for observation.

6. The quantification of cis-IEVs should be included in Figure 6E.

We apologize, but due our focus on quantifying MVB-IEVs, and the IMVs in their surroundings, we cannot accurately quantify cis-IEV wrapping from these images. The quantitative EM presented in Fig. 6B, which shows that MVB-IEVs represent ~50% of IEVs, combined with the data throughout the manuscript which shows ~50% reduction in EEV formation upon ESCRT protein depletion provides a compelling case that Cis wrapping occurs in the absence ESCRT.

7. The effects of VPS4A knockdown shown in Figures 1C, 2C and 2D don't seem to correlate with the data in Figure 6E. This needs an explanation or comment.

As mentioned in the results section corresponding to Figure 2, "Analysis of primary and secondary infection showed that depletion of VPS4A did not impact primary infection but was highly variable with regard to secondary infection".

We saw this in both Fig. 2C, 2D and 7E. Given that VPS4A depletion has no effect on MVB-IEV formation we believe this variability comes down to the variable impact of VPS4 siRNA on VPS4B expression. We have added new data (Fig. 2A) and the corresponding results addressing this issue.

The reviewer will note with the exception of 2D (2E in the revised version), which also shows a great deal of variability, none of the values recorded for VPS4A siRNA are statistically significant.

Reviewer #3 (Comments to the Authors (Required)):

Poxviruses undergo a complicated and not well-characterized membrane wrapping process as part of their maturation. This study began with a screen to identify cellular factors involved in virus egress and ultimately focused on the role of the ESCRTs in EEV formation and wrapping of IMVs at MVBs. The identification of MVBs as a potential source of poxvirus membrane as an alternative to the traditional cisterna wrapping model would be a significant finding and the data are compelling, however there are several areas that need to be addressed to solidify the interpretations and support the conclusions of the paper.

1. A major concern is the integration of the siRNA screen with the rest of the manuscript. The ESCRT components identified in the screen to affect virus egress were Tsg101, Vps4A and NEDD4. Tsg101 had already been shown to affect production of EEVs. However, no additional work was done on NEDD4 and it was VPS4B and not VPS4A that was later shown to be most important for EEV formation. It does not appear that VPS4B or the ESCRT-III components were included in the screen. If they were included, there is no discussion as to why they were not hits. If they were not included, then the significance of the screen is significantly weakened with respect to rest of the manuscript. Thus, as currently presented, it is not clear if the screen in Figure 1 is necessary and supportive of the rest of the manuscript.

We see the reviewers point with regard to the screening data not being necessary to support the follow-up data on MVB-IEV formation. However, this is the first cell-

based siRNA screen directed at identifying membrane trafficking components required for VACV spread, it confirms and extends the original TSG101 findings, led to the follow-up analysis presented in this manuscript and will serve as a valuable resource to the poxvirus community and others interested in virus exocytosis and spread. For these reasons we prefer to keep the screen in the manuscript in its present form.

In fact, it appears that Figures 1C and 2A are in direct conflict to each other. Is this a result of the difference in the time point (48 vs 72)? If so, please discuss. Based on Figure 2A, siVPS4A would not have been a hit in the initial screen as it did not result in a 25% reduction in spread.

Sorry for the ambiguity here, we briefly touched on this in the original version of the manuscript, As stated, "In this case knockdowns over 48 h, as opposed to 72 h, were used. Using the spread screen (Fig. 1A) under these conditions we found that depletion of VPS4B, but not VPS4A, reduced virus spread (Fig. 2A)". but agree this did not provide sufficient explanation.

In the siRNA screen we used a standard siRNA knockdown protocol that we have previously used for genome-scale siRNA screens (40nM siRNA/72h knockdown). Having identified VPS4A as a hit, in the follow-up we decided to include VPS4B depletion as well. When performing qRT-PCR validation we found that depletion of either VPS4A or VPS4B for 72h resulted in altered expression of the other paralog.

In an attempt to prevent this effect, and determine which protein was responsible for the spread phenotype, for subsequent experiments we established a less-harsh siRNA depletion protocol for VPS4A and B (2 subsequent 10nM siRNA transfections: 1 reverse transfection at seeding and a second at 24 h). This proved very effective for achieving specific depletion of VPS4A and VPS4B as illustrated in the qRT-PCR validation shown in Fig. 2B.

The 72h knockdown data has been added to Figure 2, and the results section extended for clarity.

2. On a related note, if VPS4B is increased 2.5-fold upon VPS4A depletion and is the more important of the two factors, why is any phenotype observed at all with VPS4A depletion in Figure 1C? Please discuss why the increase in VPS4B would not compensate.

As clarified above Fig. 1C was performed under the original screening siRNA protocol (40nM siRNA/72h knockdown) in which extended knockdown of VPS4A resulted in complete depletion of VPS4B. The loss of VPS4B is causing the spread defect in Figure 1C.

Conversely, when VPS4B levels are increased by 2.5-fold upon VPS4A depletion using the less-harsh siRNA strategy described above (Fig. 2B), an increase in EEV release and core/EEV wrapping membrane colocalization was seen. That is consistent with VPS4B, and not VPS4A, being important for MVB-IEV formation.

3. The lack of colocalization between the core and IEV wrapping machinery in the VPS4B sample of Figure 2D led to the conclusion that "VPS4B is a pro-viral factor involved in VACV EEV formation" However, the nucleoplasm of the VPS4B knockdown in Figure 2D appears dispersed, particularly in comparison to knockdown with VPS4A. How is this related to EEV formation? Could an alternative interpretation be that VPS4B disrupts another viral process upstream of wrapping, resulting in

dispersed nucleoplasm and the lack of colocalization? Knockdown of VPS4 affects many aspects of normal cell biology and additional data is needed to support the claim that VPS4B is specifically involved in EEV formation.

The position and distribution of viral replication sites (magenta signal where IMVs are formed) has little to do with IEV formation. IMVs are actively transported from replication sites to wrapping sites on microtubules, yet disruption of microtubules does not prevent wrapping but results in the formation of multiple wrapping sites. That our data shows no significant defect in IMV formation upon VPS4B depletion but a defect in MVB-IEV wrapping (assessed using multiple assays) and subsequent EEV release, and we further demonstrate that MVBs are decorated with VACV proteins involved in wrapping and that the decorated ILVs are released into the supernatant provides solid evidence that VPS4B is required for VACV EEV formation.

4. The divergent function of CHMP4B is quite striking. Given that CHMP4B is the main filament forming protein in ILV formation, which is analogous to the mechanism you are proposing for wrapping into MVBs, why would CHMP4B repress?

We agree with the reviewer as this also surprised us. The short answer is, we don't know. A wild speculation could be that it has to do with size. Perhaps, filament formation requires different CHMPs for make a 30-120nm vesicle as opposed to a 400nm "vesicle". While this is clearly an interesting finding, it is out of the scope of the current manuscript.

What is known about the isoform distribution of CHMP4 isoforms in the HeLa cells used in Figure 3? Are 4A and 4C more abundant?

We assessed the levels of CHMP4A, B and C in Hela cells by RTqPCR. The data is included below for the reviewer. In short. A is slightly more abundant than B with C being the least abundant.

This interesting but counterintuitive observation would be strengthened if repeated in CHMP4 knockdown during infection of other cell types used for poxvirus infection. Along this note, what happens to levels of the CHMP proteins during infection? Specifically, does the virus downregulate the repressive CHMP4B or upregulate CHMP4A or C? What about the other CHMPs?

While we agree with the reviewer that the CHMP4 isoform data is interesting it is out of the scope of this manuscript and rather a topic for more extensive follow-up

experimentation.

5. Please add scale bars to the insets in Figure 5.

As these figures are already quite busy, in lieu of scale bars we have added the zoom factor to the figure legends for the insets in Figs. 5 and 7.

6. In Figure 6, what happens to the total number of MVBs upon knockdown of VPS4, specifically VPS4B? While zero IEV-MVBs were reported after VPS4B knockdown (and only 4% after VPS4A knockdown), one would expect that you also would have significant reduction in MVBs since ILV formation would also be disrupted upon depleting VPS4. Thus, reporting percentage of IEV-MVBs is not informative without accounting for a difference (if any) in total MVBs.

Sorry, but this is a bit of a circular argument. The fact that we have less MVB-IEVs in the absence of VPS4B is consistent with a block in MVB formation and therefore less MVB-IEVs. As these assays were focused on assessing the presence of MVB-IEVs, the images are not suitable for making an overall determination of total MVB numbers with any certainty.

7. Importantly, what are the fate of the MVBs containing IEVs? The contents of many MVBs are destined for lysosomal degradation while others are released as extracellular content. Does the virus alter the fate of MVB containing IEVs so that the IEVs are predominantly released as EEVs? Or is a significant portion of MVB-associated IEVs degraded in lysosomes?

[Figure removed by editorial staff per authors' request]

In addition, throughout our extensive EM analysis, we have seen no evidence of virions in MVB-lysosomes. In addition, if a significant proportion of MVB-IEVs were degraded in lysosomes, one would expect depletion of ESCRT components to either have no effect (as the IMVs destined for degradation would be trapped on the MVBs anyways), or result in increased EEV production (should IMVs be 're-routed for Cisternae-based wrapping). As ESCRT depletion result in decreased EEVs, it reasons that MVB-IEVs are predominantly released as EEVs.

Perhaps this is why CHMP4B represses EEV formation. While it is clear from the data that EEVs are reduced upon knockdown of ESCRT-III and VPS4, an alternative interpretation consistent with the data in the manuscript is that MVB-associated IEVs are actually a dead end destined for degradation and that the reduction in EEVs observed with VPS4B and CHMP knockdowns are an indirect effect of disrupting another viral or cellular process.

As outlined above, I don't think this interpretation is consistent with the data.

More work is required to show that MVB-associated IEVs are released as EEVs.

We have now demonstrated that VACV MVB-IEVs are formed in two cell types, that depletion of ESCRT components results in fewer MVB-IEVs, fewer released EEVs and that abundant B5-containing extracellular vesicles (which we only observed in MVBs) are released from cells into the supernatant. We feel that collectively these results make a very strong case that MVB-IEVs are released as EEVs.

May 27, 2021

RE: Life Science Alliance Manuscript #LSA-2020-00910-TR

Prof. Jason Mercer
University of Birmingham
Institute of Microbiology and Infection
Edgbaston
Birmingham B15 2TT
United Kingdom

Dear Dr. Mercer,

Thank you for submitting your revised manuscript entitled "Vaccinia Virus Hijacks ESCRT-mediated Multivesicular Body Formation for Virus Egress". We would be happy to publish your paper in Life Science Alliance pending final revisions necessary to meet our formatting guidelines.

Please also attend to the following:

- please add ORCID ID for secondary corresponding author-they should have received instructions on how to do so
- please use the [10 author names, et al.] format in your references (i.e. limit the author names to the first 10)
- please add a callout for Figure 8 to your main manuscript text
- please revise the inset position in Figure 5C 3' so that they match the zoomed-in parts
- please add your supplementary table legend to the main manuscript text after the figure legends
- please add scale bars to Figures 1C, 5B

A. FINAL FILES:

-- High-resolution figure, supplementary figure and video files uploaded as individual files: See our detailed guidelines for preparing your production-ready images, <https://www.life-science->

alliance.org/authors

B. MANUSCRIPT ORGANIZATION AND FORMATTING:

Sincerely,

Shachi Bhatt, Ph.D.
Executive Editor
Life Science Alliance
<http://www.lsjournal.org>
Tweet @SciBhatt @LSAJournal

Reviewer #1 (Comments to the Authors (Required)):

The authors provide compelling evidence that the ESCRTs are involved in packaging IEVs into MVBs. These findings build upon a previous study in which other ESCRT proteins, ALIX and Tsg101, were found to have a role in the release of EEVs. The identification of the IEV-MVBs is certainly of interest to to the poxvirus assembly field as well as investigators studying the assembly and egress of other unrelated viruses.

The authors provide some additional data to address reviewer comments following a first round of reviews. This additional data, particularly in figure 7, strengthens the manuscript. While some important reviewer points were not adequately addressed, the manuscript is improved and these intriguing findings should be disseminated to the field.

There are no further suggestions to address the main points of the manuscript.

As with the original submission, the text is well-written and the figures are laid out in a clear manner.

June 4, 2021

RE: Life Science Alliance Manuscript #LSA-2020-00910-TRR

Prof. Jason Mercer
University of Birmingham
Institute of Microbiology and Infection
Edgbaston
Birmingham B15 2TT
United Kingdom

Dear Dr. Mercer,

Thank you for submitting your Research Article entitled "Vaccinia Virus Hijacks ESCRT-mediated Multivesicular Body Formation for Virus Egress". It is a pleasure to let you know that your manuscript is now accepted for publication in Life Science Alliance. Congratulations on this interesting work.

DISTRIBUTION OF MATERIALS:

Again, congratulations on a very nice paper. I hope you found the review process to be constructive and are pleased with how the manuscript was handled editorially. We look forward to future exciting submissions from your lab.

Sincerely,
